# Transmission patterns of tick-borne pathogens among birds and rodents in a forested park in southeastern Canada

**Ariane Dumas**[1,2]*, **Catherine Bouchard**[2,3], **Antonia Dibernardo**[4], **Pierre Drapeau**[5], **L. Robbin Lindsay**[4], **Nicholas H. Ogden**[2,3], **Patrick A. Leighton**[1,2]

**1** Department of Pathology and Microbiology, Faculty of Veterinary Medicine, Université de Montréal, Saint-Hyacinthe, Quebec, Canada, **2** Epidemiology of Zoonoses and Public Health Research Unit (GREZOSP), Faculty of Veterinary Medicine, Université de Montréal, Saint-Hyacinthe, Quebec, Canada, **3** Public Health Risk Sciences Division, National Microbiology Laboratory, Public Health Agency of Canada, Saint-Hyacinthe, Quebec, Canada, **4** One Health (Formerly Zoonotic Diseases and Special Pathogens) Division, National Microbiology Laboratory, Public Health Agency of Canada, Winnipeg, Manitoba, Canada, **5** Department of Biological Sciences, Centre for Forest Research, Université du Québec à Montréal, Montreal, Quebec, Canada

* ariane.dumas@umontreal.ca

**Data Availability Statement:** All relevant data are within the manuscript and its Supporting information files.

## Abstract

*Ixodes scapularis* ticks are expanding their range in parts of northeastern North America, bringing with them pathogens of public health concern. While rodents like the white-footed mouse, *Peromyscus leucopus*, are considered the primary reservoir of many emerging tick-borne pathogens, the contribution of birds, as alternative hosts and reservoirs, to local transmission cycles has not yet been firmly established. From 2016 to 2018, we collected host-seeking ticks and examined rodent and bird hosts for ticks at 48 sites in a park where black-legged ticks are established in Quebec, Canada, in order to characterize the distribution of pathogens in ticks and mammalian and avian hosts. We found nearly one third of captured birds (n = 849) and 70% of small mammals (n = 694) were infested with *I. scapularis*. Five bird and three mammal species transmitted *Borrelia burgdorferi* to feeding larvae (n larvae tested = 2257) and we estimated that about one fifth of the *B. burgdorferi*-infected questing nymphs in the park acquired their infection from birds, the remaining being attributable to mice. Ground-foraging bird species were more parasitized than other birds, and species that inhabited open habitat were more frequently infested and were more likely to transmit *B. burgdorferi* to larval ticks feeding upon them. Female birds were more likely to transmit infection than males, without age differentiation, whereas in mice, adult males were more likely to transmit infection than juveniles and females. We also detected *Borrelia miyamotoi* in larvae collected from birds, and *Anaplasma phagocytophilum* from a larva collected from a white-footed mouse. This study highlights the importance of characterising the reservoir potential of alternative reservoir hosts and to quantify their contribution to transmission dynamics in different species assemblages. This information is key to identifying the most effective host-targeted risk mitigation actions.

**Funding:** Our work was funded by the Public Health Agency of Canada (PHAC) and a Natural Sciences and Engineering Research Council of Canada (NSERC; https://www.nserc-crsng.gc.ca/) Discovery Grant to P.A.L. (#03793–2014), with fellowship support for A.D. provided by NSERC, the Fonds de recherche Nature et Technology du Québec (FRQNT; https://frq.gouv.qc.ca/) and the Université de Montréal. The funders had no role in study design, data collection and analysis, decision to publish, or preparation of the present manuscript.

**Competing interests:** The authors have declared that no competing interests exist.

## Introduction

Tick-borne pathogens are emerging in many parts of North America, with warming climate and other environmental changes favoring range expansion of ticks and their pathogens into new regions [1]. In Canada, the blacklegged tick, *Ixodes scapularis*, is establishing new populations at higher latitudes bringing with it the risk of transmission of tick-borne pathogens to resident human populations. Among these pathogens, the most common is *Borrelia burgdorferi*, the agent of Lyme disease, but others are emerging and represent public health concerns including the agents of Anaplasmosis, Babesiosis, Powassan disease and *Borrelia miyamotoi* disease [1]. In order to establish enzootic transmission cycles, these pathogens depend on the interaction between tick vectors and vertebrate hosts that are capable of serving as reservoirs. The reservoir competence of a given host species is determined by its capacity to acquire and transmit the pathogen to another vertebrate (in this case via the tick vector), thus perpetuating the pathogen in the environment [2].

Multiple species of small mammals serve as reservoirs for emerging tick-borne pathogens in Canada [1]. Mouse species, chipmunks, squirrels, and shrews has been recognised as reservoirs of the agents of anaplasmosis and Lyme disease [3]. In particular, white-footed mice, *Peromyscus leucopus* are a reservoir for the agents of several tick-borne diseases (e.g. babesiosis, *Borrelia miyamotoi* disease, Powassan encephalitis) and is considered as the most important reservoir host for *Borrelia burgdorferi*, the bacterial cause of Lyme disease [3]. The high level of reservoir competence exhibited by white-footed micefor these pathogens has been attributed to multiple factors. First, this species is very efficient in transmitting infection to ticks. For example, a xenodiagnosis experiment, conducted on small and medium hosts captured in Connecticut forests, highlighted the highest reservoir competency of white-footed mice compared to the other species, using RT-PCR to quantify the number of *B. burgdorferi* bacteria per tick collected after bloodmeal completion and moulting [2]. Also, white-footed mice can remain infected and infectious for ticks life-long for some bacterial strains (reviewed by [4]). Finally, the ubiquity of white-footed mice in natural and anthropized ecosystems in northeastern North America is also thought to accentuate its importance as a reservoir of tick-borne pathogens in this region [5–7]. In numerous studies, birds have been identified as long-distance dispersers of ticks and tick-borne pathogens along their seasonal migration routes [8–11]. In Europe, where different genospecies of *B. burgdorferi* sensu lato circulate, the transmission dynamics of some genospecies (particularly *B. garinii*) are driven by avian reservoirs, and others (*B. afzellii*) by rodents [12]. In North America, multiple bird species are capable of acquiring and efficiently transmitting the species-generalist genospecies *B. burgdorferi* sensu stricto to feeding ticks [13–17]. Despite these findings, the relative lower density of birds compared to that of rodents is thought to limit their overall contribution as reservoirs [18, 19].

At the scale of an ecological community, the contribution of a species to the transmission dynamics of a tick-borne pathogen is determined by the proportion of ticks that acquired the pathogen from individuals of that species, and thus depends on the frequency with which they are fed upon by ticks, their level of infectivity and their abundance in the ecosystem [20]. In a modeling study, Giardina et al. [21] compared the proportion of nymphs that acquired their infection from birds versus rodents and concluded that birds had a negligible contribution to the overall transmission dynamics of *B. burgdorferi* s.s. However, empirical studies and data on the role of birds in local transmission cycles of *B. burgdorferi* s.s. and other tick-borne pathogens in North America are limited [22]. Greater knowledge of the role of birds is therefore needed to assess the risk of Lyme disease, and the potential impact of control measures. In particular, control strategies in which hosts are vaccinated or treated with acaricide are gaining research attention as promising avenues for the reduction of the environmental risk of tick-

borne disease [23–25]. Identifying the main reservoirs and their relative importance will be critical to ensuring the efficacy of host-targeted control efforts.

The natural history of tick-borne pathogens other than *B. burgdorferi* has received even less research attention. The reservoir competence of American Robins (*Turdus migratorius*) and Gray Catbirds (*Dumetella carolinensis*) for *A. phagocytophilum* was tested in laboratory assays conducted by Johnston et al. [26] and the results suggested that these species were capable of transmitting the pathogen but were unlikely to play a significant role in transmission in a natural context. Two strains of *A. phagocytophilum* circulate in North America, one that is pathogenic to humans (Ap-ha) with rodents (e.g., white-footed mice, chipmunks, squirrels, and shrews) as the primary reservoir, and the other that is non-pathogenic to humans (Ap-variant 1) with white-tailed deer as the primary reservoir [27]. The reservoirs of *B. miyamotoi* are less well known. The white-footed mouse is a competent reservoir in eastern North America [28, 29] and other species, including birds, may also act as reservoirs [30]. In a study conducted in the Netherlands, Wagemakers et al. [31] found similar rates of *B. miyamotoi* infection in biopsies from rodents and birds, while in the USA, a study revealed high prevalence of *B. miyamotoi* in wild turkeys [32]. However, further studies are needed to determine the reservoir hosts of *B. miyamotoi*, which is a challenging task because larvae can acquire infection transovarially from infected adult females and thus testing feeding larvae from hosts may not mean that the host was infected or acting as a reservoir [33].

In this study, we collected ticks and hosts in a forest with newly endemic Lyme disease risk in order to: i) characterize the distribution of three emerging zoonotic pathogens (*B. burgdorferi*, *B. miyamotoi*, and *A. phagocytophylum*) in ticks and tick hosts; ii) compare the contribution of avian hosts to *B. burgdorferi* transmission to that of white-footed mice; and iii) determine risk factors for tick infestation and *B. burgdorferi* infectivity among hosts. By comparing, for the first time in North America, the relative roles of breeding birds and rodents in maintaining enzootic cycles of emerging tick-borne pathogens including *B. burgdorferi*, we aim to provide a more complete picture of local host-vector-pathogen dynamics. This will improve understanding of the ecology of tick-borne pathogens and facilitate accurate risk assessment and development of effective host-targeted control strategies.

## Materials and methods

### Field

This study was conducted at Mont Saint-Bruno National Park (Quebec), during the summers of 2016 to 2018 (Fig 1). Mont Saint-Bruno is a forested hill (elevation: 218 m above sea level) where stands of deciduous tree species dominate, mainly sugar maple (*Acer saccharum*), American beech (*Fagus grandifolia*) and red oak (*Quercus rubra*). There are also several open areas attributable to man-made modifications throughout the site's history, including former orchards, mills and cottage lots surrounded by gardens. This diverse habitat constitutes a biodiversity island of nearly 9 km$^2$, located in the middle of an urban and agricultural plain. Located just outside of the city of Montréal, it is a popular destination for hikes, and attracts approximately one million visitors each year. A local population of blacklegged ticks, *I. scapularis*, as well as *B. burgdorferi* transmission cycles, has been established at this location for several years [34, 35].

We collected questing ticks by drag sampling over a total area of 260 m$^2$ per visit, at 32 sites distributed throughout the forested areas of the park (Fig 1), once a month between May and October of each year of the study, as described in [36]. In 2018, we added two visits (June and July) at open-fields where the birds were captured (see below), in order to characterize the exposure of captured birds to ticks in their habitat.

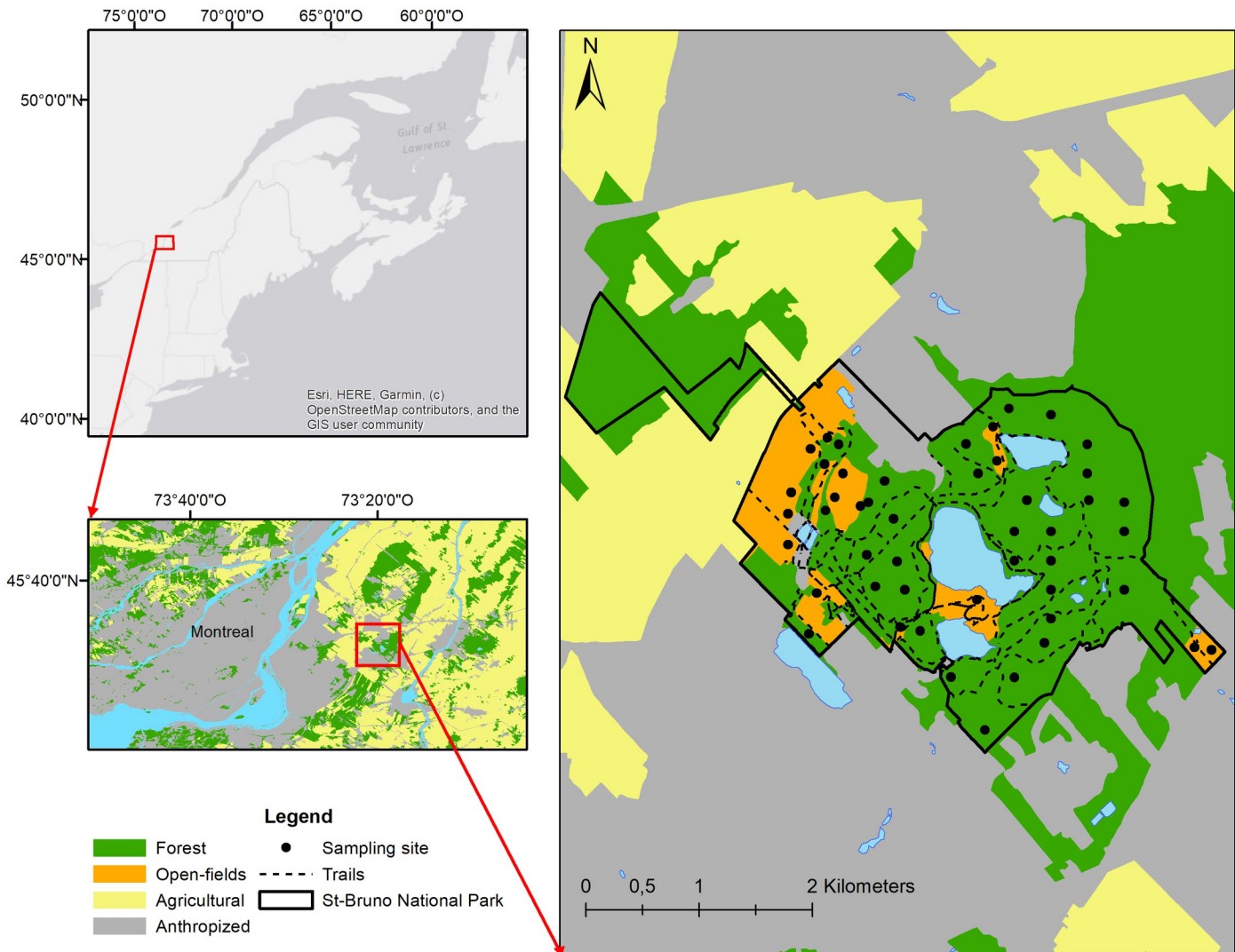

**Fig 1. The location of the study site and sampling localities in Quebec Canada.** For illustrative purposes, the map is an aggregated representation of habitat types and land use categories based on ecoforestry inventory data obtained from the Ministère des Forêts, de la Faune et des Parcs du Québec (https://www.foretouverte.gouv.qc.ca/). The base map is from OpenStreetMap and OpenStreetMap Foundation.

We aimed to sample hosts during the main period of the year when they are abundant. These periods also covered the main period of immature tick activity at this site [36]. Small mammals were trapped at each of the 32 forest sites, over a five-week period in July for three consecutive years (2016–2018). Sites were sampled on a rotating basis, with an effort of four capture days per site per year. At each site, 30 Sherman™ live traps were placed in 100 x 40 m rectangular grids, baited with peanut butter and apples and checked twice daily. Captured rodents were anesthetized with isoflurane, each animal was then weighed, and the sex and species of each host was determined. For white-footed mice, the relative age of each individual was determined by a weight classification [37]. All captured rodents were marked individually with subcutaneous PIT tags (model HPT8, BioMark™), and the entire body surface of each host was examined visually for ticks. All attached ticks were removed using fine tipped forceps and ticks were placed into 1.5 ml tubes containing 70% ethanol. A 3 mm ear punch biopsy was

collected from each rodent and placed in a 1.5 ml tube containing 70% ethanol and kept at room temperature until laboratory testing.

Birds were collected during 2017 and 2018 using four 12 meter-long mist nets (height = 2.6 m). To maximize diversity of captured birds, we collected birds at 16 open-field sites (Fig 1) in addition to a subset of 16 forest sites. We carried out two capture periods each year: the first one in early summer, during the breeding period, and the second one at the end of the summer, the post-breeding period during which family groups start to disperse [38]. In each trapping period, we censused and captured birds five days per week for five weeks (see below). At each site visit, we first conducted point counts using observation and characteristic bird song during the breeding period. Nets were opened from 5 to 11 a.m., unless adverse meteorological conditions such as heavy rain or extreme heat threatening the birds' well-being occurred. In order to maximise capture success, we lured birds to the nets using playback recordings of conspecific songs of detected species at each site. Studies have found that in addition to increasing the capture success of target species, the use of playbacks while using mist nests could also favor capture [39, 40], or response [41] of other species, not targeted by the playback. We checked the nets and removed the birds as quickly as possible, with a maximum delay of 30 minutes. The bander's grip [42] was used in the majority of cases for bird handling and great care was taken with plumage to avoid damage to feathers or unnecessary displacement. Captured birds were processed according to the order of capture and size of birds, with priority given to smaller birds. Species and sex of captured birds were determined and relative age (juveniles corresponding to hatch-year birds (HY) and adults second-year birds (SY) and after second-year (ASY) birds) of each capture was estimated according to molt and plumage [43]. Each bird was weighed, marked individually with unique bands, and examined visually for ticks. Attached ticks and biopsies were collected as described for rodents except a skin biopsy was collected from the prepatagial membrane of each bird using 2 mm a disposable biopsy punch. Biopsy samples were placed in tubes containing saline buffered with glycerin, preserved on ice in the field and frozen in the laboratory [44]. Any bird showing early signs of hypothermia were placed in coolers containing air-activated heat packs until they recovered, after which they were released.

Every captured animal was released at the site of capture following manipulations. All animal manipulation protocols were approved by Université de Montréal ethics committee (certificates no: 16-Rech-1837, 17-Rech-1837 and 18-Rech-1837). Bird captures were authorized by Canadian Wildlife Service (permit no: 10739D) and small mammals captures were authorized by Quebec Ministères des Forêts, de la Faune et des Parcs (certificates no: 2016SF2063R16, 20170508222316SF and 2018425241416-SF).

All ticks found on animals or removed from drag cloths were sent to the National Microbiology Laboratory for species identification and pathogen testing.

## Diagnostic testing for tick-borne pathogens

All ticks were identified to species using taxonomic keys [45–47]. For ticks collected by drag sampling, only a subset of up to 30 ticks of each stage per location per year were tested. For ticks removed from hosts, we tested all larvae, and a subset of up to 30 nymphs per infested host species for nymphs collected from birds. Nymphs removed from small mammals were not tested. Level of blood engorgement was determined visually and scored as: partially to fully engorged, slightly engorged, and unfed. Unfed larvae from the same individual hosts were pooled together for testing (pool sizes ranged from 2 to 27 ticks). The samples were processed within approximately one year from the date of collection. Extraction of DNA from ticks and biopsy samples was performed according to the manufacturer's protocol using

QIAGEN®DNeasy®96 Tissue kits (QIAGEN Inc., Mississauga, ON, Canada). Real-time PCR was used to detect *B. burgdorferi*, *B. miyamotoi* and *A. phagocytophylum*. Briefly, extracted DNA was screened using a duplex real-time PCR assay targeting the *23S* and *msp2* genes of *Borrelia* spp. and *A. phagocytophilum*, respectively [48]. *Borrelia*-positive samples were subsequently tested for *B. burgdorferi* and *B. miyamotoi* using a second *ospA*/*flab* duplex assay [49]. Biopsies from birds were taken only in 2017, and since no positive samples were detected (see Discussion), we then abandoned this technique in subsequent years. DNA collected from biopsies of *Peromyscus* spp. was also screened using species-specific primers (*CO1*) in multiplex PCR to differentiate between the sympatric and morphologically similar deer mice (*Peromyscus maniculatus*) and white-footed mice (*P. leucopus*) [50].

## Statistical analysis

**Pathogen prevalence in *I. scapularis* ticks.**   All analyses were carried out using R v. 4.0.3 [51]. We compared the prevalence of the three pathogens tested in *I. scapularis* ticks using Chi-square tests, according to their origin: questing nymphs in the environment, or feeding larvae on a host. We obtained the proportion of feeding larvae that acquired *B. burgdorferi* for each host species by combining results of ticks tested individually and pools of unfed ticks in a weighted (by the number of ticks) mean. We estimated the proportion of infected ticks in each pool using the Maximum-Likelihood Estimation Infection Ratio method [52].

**Host risk factors for tick infestation and *B. burgdorferi* infectivity.**   To analyze the risk factors for tick infestation density and the probability of transmission of *B. burgdorferi* to feeding ticks, we constructed generalized mixed models with the package *glmmTMB*, version 1.1.1 [53]. The individual host was used as the analytical level. Infestation density refers to the number of ticks per host examined in our sample, including uninfested hosts [54]. Vertical transmission of *B. burgdorferi* does not occur in *I. scapularis* [33]. We therefore used infection in feeding larvae as an indicator that hosts could infect ticks [6, 55], and quantified infectivity as the proportion of larvae infected after feeding on a given host. We used negative binomial distributions for the infestation models, and a binomial distribution with a logit link for the infectivity models. The dependent variables for these two models were: 1) the number of larvae per individual and 2) the proportion of infected versus uninfected larvae. As explanatory variables, we tested a set of variables that could impact the probability of contact between hosts and ticks, and the probability of pathogen transmission between them, as identified in other studies (Table 1). Intrinsic individual factors such as age and sex of animals are thought to impact the risk of infestation and infectivity in mice [35, 56] and other factors such as weight and body condition in birds [8]. Given the wide diversity of avian species present at the study site and in order to explore ecological factors associated with different levels of risk between species, we also included in the bird models life history traits that are thought to modify the probability of contact with ticks, namely nest placement, foraging behavior, and preferred habitat of the species [22]. The classification of life history traits by species was based on that of the Cornell Lab of Ornithology [57]. Families were added to the bird infectivity model to account for the fact that different taxonomic groups may have developed different immune responses to tick-borne infections [15]. Next, to account for the spatial and temporal dependence that our infestation data may have, we added a capture site ID as a random intercept term, as well as the average predicted density of questing larvae in the environment on the calendar day of capture as a fixed effect term, which corrects for seasonal variability attributable to tick phenology [36]. This correction is necessary because the infestation density of hosts is driven by the contact rate with vectors, which in turns depends on the density of ticks in the environment [58]. Finally, since *B. burgdorferi* multiplies in the tick as it ingests blood from the infected host

**Table 1. Description of variables tested in host infestation and infectivity models.**

| Type | Variable | Infestation | | Infectivity | | Unit | Expected relationship |
|---|---|---|---|---|---|---|---|
| | | Birds | Mice | Birds | Mice | | |
| Intrinsic factors | Age | x | x | x | x | Juvenile / Adult | A > J |
| | Sex | x | x | x | x | Male / Female | M > F |
| | Weight | x | x | x | x | Grams | + |
| | Fat score | x | | x | | 0–4 | - |
| Taxonomic | Family | | | x | | 8 bird families | Different immune responses |
| Life-history traits | Nest location | x | | x | | Low (ground) / High (building, cavity, shrub or tree) | L > H |
| | Feeding behavior | x | | x | | Bark Forager, foliage gleaner, ground forager, in flight (flycatching, or hovering) | GF > BF > FG > F |
| | Main habitat | x | | x | | Forest / Open-fields (open woodland, scrub, grassland and marshes) | F > O |
| Corrections | Questing larvae by Julian day and year | x | x | | | Number of ticks | + |
| | Year | | | x | x | 2017, 2018 | NA |
| | Site ID | x | x | x | x | 48 sites | Random intercept |
| | Engorged feeding larvae | | | x | x | Number of ticks | + |

[59], we added the proportion of feeding larvae on each host that were engorged as a predictor in the infectivity models. Prior to model building, we verified that there was no collinearity between the explanatory variables, using a variance inflation factor (VIF) threshold of 3 [60]. The final models were selected by backward stepwise elimination. For categorical variables, we compared the levels with the Tukey adjustment for multiple comparisons with a significance level of p = 0.05. We finally confirmed the absence of residual spatial autocorrelation in the final models using Moran's I tests [61], implemented with package *ape*, version 5.5 [62]. The fit of the models was verified graphically and with the diagnostic tests of the *DHARMa* package, version 0.4.1 [63].

**Contribution of hosts to *B. burgdorferi* transmission.** We estimated the reservoir potential (RP) of each species for which we detected *B. burgdorferi* according to Eq 1 [6], where the parameter *Infestation$_i$* represents the average larval infestation density of individuals of species *i*, *Infectivity$_i$* the average proportion of larvae infected after feeding on individuals of a given species *i*, and *Host density$_i$* the estimated population density of the species *i* in our study area (number of individuals per hectare).

$$RP_i = Infestation_i * Infectivity_i * Host\ density_i. \tag{1}$$

Capture periods for birds and mice did not overlap, so to compare between birds and mice as hosts for larvae, we adjusted observed tick abundance on hosts according to the seasonality of larvae in the environment [35, 64]. To do so, we first constructed a model for individual infestation density in reservoir birds and another for white-footed mice (all *Peromyscus* spp. captured were determined to be white-footed mouse and not deer mouse by PCR). Variables included in these models were age, sex and weight of the animal. For birds, we took into account the non-independence of data from individuals of the same species, but grouped them by family, since the highly variable number of observations per species prevented us from using this factor in the models, generating convergence problems. To these variables we added a predicted value for larval density in the environment on the day of capture according to smoothed seasonality curves as described in Dumas et al. [36]. Capture site was included as a random effect to correct for spatial dependence and repeated measurements at the site [60].

We excluded recaptures in the calculation of infestation density to avoid a possible biais associated with the lack of independence between tick abundance on hosts and their capture status (first capture versus recapture). Ticks were removed from animals as part of processing and the same rodents could be recaptured within the same week. This was not true for birds though because successive capture periods at the same site were spaced a few weeks apart. For both models, we performed backward stepwise model selection and a negative binomial distribution using the package *glmmTMB*, version 1.1.1 [53]. We then calculated the values predicted by the model for a reference date (the 200th day since January 1st), which corresponds to the middle of the capture period of both groups, as the corrected value for calculation of the species reservoir potential.

White-footed mice were the only small mammal species for which we obtained sufficient captures in order to estimate population density. Raccoons in our study area frequently disturbed rodent traps leading to unequal sampling efforts between sites and visits. We therefore used a spatially explicit capture-recapture (package *secr*, version 4.4.1) method, allowing us to account for uneven sampling effort [65]. Disturbed traps were considered to be active at 50%, as the trap may have been sprung at any time during its deployment period. We set the influence distance of the traps to 4 times the spatial scale parameter (representing the tendency for detections to be localised), with a half-normal detection function (detection probability declines with increasing distance from the traps [66]). We estimated density for each site and year then used the average of these values as the overall density of the species in the study area.

We estimated bird density from point count surveys, which we conducted twice each year at each of the sites, during the breeding season. The point counts were all conducted between 5:00 and 9:30 am, on days without rain and with winds below 25 kilometres per hour. At each visit, three replicate blocks of five minutes were performed to maximize detection. Every bird heard or seen was directly identified by trained observers. We used the maximum number of male individuals in a 100-meter radius at any 5 minute listening block as a baseline value for each species, to which we added juvenile and female density estimates, assuming a 1:1 male-to-female ratio [21] and a juveline-to-adult ratio calculated using capture data from the post-fledging period (August). We then estimated abundance of every species for the entire study area by calculating the mean number of individuals present per site and habitat type (forest vs. open fields) and weighting this value by the proportion of the study area covered by each habitat. Finally, these estimates were adjusted to account for variable detectability of bird species, which is associated with the average frequency at which individuals of a species produce sound signals, the singing rate (SR), and the average distance at which their song can be detected, the detection distance (DD) [67]. Although these two parameters can be estimated directly from survey data, we detected too few individuals of each species to perform the analysis and we instead used published estimates [67] based on a time-removal model for SR [68] and a distance-sampling model [69] for DD. SR defines the probability that individuals of a given species are available to be perceived, if present (availability, *p*, Eq 2), and DD defines the probability that an individual of the species will be detected if it produces a sound signal (perceptibility, *q*, Eq 3). The probability of detection is the product of these two components (*pq*) [70].

$$p = 1 - \exp(-t * SR) \qquad (2)$$

$$q = \frac{DD^2 \, (1 - exp(-r^2/DD^2))}{r^2} \qquad (3)$$

where $t$ is the duration of the listening period in minutes, $SR$ is the singing rate of the species,

$$p = 1 - \exp(-t * SR)$$

$DD$ is the detection distance of the species and $r$ is the radius of the listening area in meters.

We compared the relative contribution of each species and family by calculating their relative reservoir potential (RRP) according to Eq 4 [6] with the numerator being the RP of the species or the sum of the RPs of the family and the denominator being the sum of the RPs of all species considered.

$$RRP = \frac{RP_i}{\sum s\,(RP_i)} \tag{4}$$

## Results

### Field sampling of ticks and hosts

We collected 25,150 larvae, 4,177 nymphs and 232 adult blacklegged ticks by drag sampling. The mean density of nymphs (DON) was 3.2/100 m$^2$ (s.d. 4.8) and ranged from 0 to 42.7 nymphs/100 m$^2$.

Small mammal trapping resulted in the collection of 665 mice (*P. leucopus*), 13 Eastern chipmunks (*Tamias striatus*), 15 Northern short-tailed shrew (*Blarina brevicauda*) and one Red-backed vole (*Myodes gapperi*). Of these, 249 mice and 18 chipmunks were recaptured at least once. We found 470 (70.68%) mice, 12 (92.31%) chipmunks and 2 (13.33%) shrews infested with at least one tick. Ticks were not found on the only vole captured. The abundance of larvae and nymphs per hosts were, respectively, 2.32 and 0.17 for mice, 2.4 and 5.77 for chipmunks and 0.23 and 0 for shrews. Ticks collected from small mammals were predominantly attached to the ears.

We captured 849 birds belonging to 50 different species, and 78 individuals were recaptured at least once. Birds from 8 families accounted for 86% of all captures (from the most to the least common these were: *Parulidae*, *Passerellidae*, *Turdidae*, *Fringillidae*, *Paridae*, *Picidae*, *Vireonidae* and *Tyrannidae*, Table 2). Ticks were found on 245 (28.86%) birds, with the majority of these ticks removed from members of the *Passerellidae* (37.41%), *Turdidae* (31.11%) and *Parulidae* (17.04%) families. Among all the bird species captured, the abundance of larvae and nymphs was 0.50 and 0.65 ticks per individual, respectively (Table 2). Two birds carried *Ixodes muris* nymphs, a Song Sparrow (*Melospiza melodia*) with two ticks and a White-throated Sparrow (*Zonotrichia albicollis*) with one. Ticks collected from birds were predominantly attached around the eyes and beak.

### Tick-borne pathogens detected in hosts tissues

We tested 625 biopsies from mice and 212 (33.92%) were positive for *B. burgdorferi*, 3 (0.48%) for *B. miyamotoi* and none for *A. phagocytophilum*. We tested 13 chipmunk biopsies and 11 (84.62%) were positive for *B. burgdorferi*, 2 (15.38%) for *B. miyamotoi* and 1 (7.69%) for *A. phagocytophilum*. The detection of *B. burgdorferi* from biopsy samples and from feeding larvae collected from the same individuals showed good agreement (Cohen's kappa = 0.67, p < 0.01).

Pathogens were not detected in any of the bird biopsies (n = 262).

### Pathogen prevalence in *I. scapularis* ticks

We tested 2,210 questing nymphs (63.52% of the total questing nymphs collected) and found an overall prevalence of 9.10% (range: 7.98–10.55%) for *B. burgdorferi* across the three years of

**Table 2. Summary of *Ixodes scapularis* infestation and *Borrelia burgdorferi*, *Borrelia miyamotoi* and *Anaplasma phagocytophilum* infection detected in ticks feeding on breeding birds captured and censused at Mont-Saint-Bruno National Park in 2017 and 2018.**

| | Birds | | | | | | Feeding larvae | | | | | | Feeding nymphs | | | | |
|---|---|---|---|---|---|---|---|---|---|---|---|---|---|---|---|---|---|
| Species | No. captured | Density (ind/ha) | Infestation prevalence | No. ticks collected | Infestation intensity[1] | | Mean infectivity[2] | Pos. Bb[3] (n) | MLE IR pools[4] | No. ticks in pools | Pos. Bm[5] (n) | Pos. Ap[6] (n) | No. ticks collected | Tested (n) | Pos. Bb (n) | Pos. Bm (n) | Pos. Ap (n) |
| **Birds that have infected *I. scapularis* feeding larvae with *B. burgdorferi*** | | | | | | | | | | | | | | | | | |
| Chipping Sparrow | 18 | 0.57 | 0.33 | 19 | 3.17 | | 0.11 | 17 | 0 | 2 | 0 | 0 | 28 | 26 | 1 | 0 | 2 |
| Hermit Thrush | 39 | 1.57 | 0.74 | 75 | 2.59 | | 0.19 | 61 | 0.08 | 14 | 1 | 0 | 141 | 30 | 6 | 0 | 3 |
| House Wren | 2 | 0.19 | 0.50 | 10 | 10.00 | | 0.20 | 5 | 0 | 5 | 0 | 0 | 2 | 2 | 0 | 0 | 0 |
| Dark-eyed Junco | 3 | 0.27 | 0.33 | 1 | 1.00 | | 1.00 | 1 | - | 0 | 0 | 0 | 4 | 2 | 1 | 0 | 0 |
| Song Sparrow | 107 | 1.16 | 0.31 | 181 | 5.48 | | 0.20 | 99 | 0.09 | 82 | 1 | 0 | 296 | 34 | 4 | 1 | 0 |
| **Subtotal** | **169** | **3.77** | **0.44** | **286** | **4.45** | | **0.34** | **183** | **0.08** | **103** | **2** | **0** | **471** | **94** | **12** | **1** | **5** |
| **Birds that were infested with *I. scapularis* but that did not transmit *B. burgdorferi* infection** | | | | | | | | | | | | | | | | | |
| American Goldfinch | 86 | 9.16 | 0.01 | 1 | 1.00 | | 0.00 | 0 | 0.00 | 0 | 0 | 0 | 2 | 2 | 1 | 0 | 0 |
| American Redstart | 24 | 6.93 | 0.04 | 1 | 1.00 | | 0.00 | 0 | 0.00 | 0 | 0 | 0 | 4 | 4 | 1 | 0 | 0 |
| American Robin | 1 | 1.19 | 0.00 | 0 | 0.00 | | - | - | - | - | - | - | 2 | 1 | 1 | 0 | 0 |
| Black and white Warbler | 10 | - | 0.30 | 3 | 1.00 | | 0.00 | 0 | 0.00 | 0 | 0 | 0 | 8 | 7 | 1 | 0 | 0 |
| Black-billed Cuckoo | 2 | - | 0.00 | 0 | 0.00 | | - | - | - | - | - | - | 1 | 1 | 0 | 0 | 0 |
| Black-capped Chickadee | 74 | 9.09 | 0.05 | 10 | 2.50 | | 0.00 | 0 | 0.00 | 0 | 0 | 0 | 3 | 3 | 1 | 0 | 0 |
| Brown-headed Cowbird | 4 | 0.27 | 0.00 | 0 | 0.00 | | - | - | - | - | - | - | 1 | 1 | 1 | 0 | 0 |
| Blue Jay | 1 | 1.01 | 1.00 | 6 | 6.00 | | 0.00 | 0 | 0.00 | 0 | 0 | 0 | 1 | 1 | 1 | 0 | 0 |
| Cedar Waxwing | 25 | 11.17 | 0.00 | 0 | 0.00 | | - | - | - | - | - | - | 11 | 10 | 1 | 0 | 3 |
| Common Yellowthroat | 28 | 0.93 | 0.11 | 10 | 3.33 | | 0.00 | 0 | 0.00 | 0 | 0 | 0 | 25 | 24 | 2 | 0 | 2 |
| Eastern Phoebe | 15 | 0.32 | 0.00 | 0 | 0.00 | | - | - | - | - | - | - | 1 | 1 | 1 | 0 | 0 |
| Gray Catbird | 26 | - | 0.00 | 0 | 0.00 | | - | - | - | - | - | - | 4 | 4 | 1 | 0 | 1 |
| Indigo Bunting | 13 | 0.62 | 0.08 | 1 | 1.00 | | 0.00 | 0 | 0.00 | 0 | 0 | 0 | 7 | 7 | 0 | 0 | 2 |
| Northern cardinal | 8 | - | 0.00 | 0 | 0.00 | | - | - | - | - | - | - | 4 | 4 | 2 | 0 | 0 |
| Ovenbird | 49 | 3.17 | 0.31 | 39 | 2.60 | | 0.00 | 0 | 0.00 | 0 | 0 | 0 | 10 | 9 | 0 | 0 | 0 |
| Pileated Woodpecker | 3 | 1.43 | 0.00 | 0 | 0.00 | | - | - | - | - | - | - | 1 | 1 | 0 | 0 | 0 |
| Rose-breasted Grosbeak | 6 | 0.86 | 0.17 | 2 | 2.00 | | 0.00 | 0 | 0.00 | 0 | 0 | 0 | 1 | 1 | 0 | 0 | 0 |
| Red-eyed Vireo | 54 | 4.92 | 0.02 | 1 | 1.00 | | 0.00 | 0 | 0.00 | 0 | 0 | 0 | 2 | 2 | 0 | 0 | 0 |
| Scarlet Tanager | 2 | 2.08 | 0.50 | 1 | 1.00 | | 0.00 | 0 | 0.00 | 0 | 0 | 0 | 3 | 3 | 2 | 0 | 0 |
| Veery | 46 | 2.36 | 0.20 | 12 | 1.33 | | 0.00 | 0 | 0.00 | 0 | 1 | 0 | 57 | 31 | 3 | 0 | 6 |

*(Continued)*

**Table 2.** (Continued)

| Species | Birds | | | | | Feeding larvae | | | | | | | Feeding nymphs | | | |
|---|---|---|---|---|---|---|---|---|---|---|---|---|---|---|---|---|
| | No. captured | Density (ind/ha) | Infestation prevalence | No. ticks collected | Infestation intensity[1] | Mean infectivity[2] | Pos. Bb[3] (n) | MLE IR pools[4] | No. ticks in pools | Pos. Bm[5] (n) | Pos. Ap[6] (n) | No. ticks collected | Tested (n) | Pos. Bb (n) | Pos. Bm (n) | Pos. Ap (n) |
| White-breasted Nuthatch | 28 | 2.60 | 0.11 | 6 | 2.00 | 0.00 | 0 | 0.00 | 0 | 0 | 0 | 6 | 6 | 1 | 0 | 0 |
| Wood Thrush | 17 | 0.72 | 0.18 | 11 | 3.67 | 0.00 | 0 | 0.00 | 0 | 0 | 0 | 20 | 18 | 0 | 1 | 0 |
| White-throated Sparrow | 5 | 0.11 | 0.00 | 0 | 0.00 | - | - | - | - | - | - | 3 | 3 | 1 | 0 | 1 |
| **Subtotal** | **527** | **58.92** | **0.13** | **104** | **1.28** | **0.00** | **0** | **0.00** | **0** | **1** | **0** | **177** | **144** | **21** | **1** | **15** |
| **Birds not infested with *I. scapularis*** | | | | | | | | | | | | | | | | |
| Blue-headed Vireo | 1 | 0.67 | 0.00 | 0 | 0.00 | - | - | - | - | - | - | 0 | - | - | - | - |
| Blackburnian Warbler | 6 | 1.08 | 0.00 | 0 | 0.00 | - | - | - | - | - | - | 0 | - | - | - | - |
| Brown Creeper | 1 | | 0.00 | 0 | 0.00 | - | - | - | - | - | - | 0 | - | - | - | - |
| Black-throated Green Warbler | 25 | 4.90 | 0.00 | 0 | 0.00 | - | - | - | - | - | - | 0 | - | - | - | - |
| Chestnut-sided Warbler | 4 | 0.19 | 0.00 | 0 | 0.00 | - | - | - | - | - | - | 0 | - | - | - | - |
| Downy Woodpecker | 24 | 2.95 | 0.00 | 0 | 0.00 | - | - | - | - | - | - | 0 | - | - | - | - |
| Eastern Wood-Pewee | 14 | 2.46 | 0.00 | 0 | 0.00 | - | - | - | - | - | - | 0 | - | - | - | - |
| Field Sparrow | 1 | - | 0.00 | 0 | 0.00 | - | - | - | - | - | - | 0 | - | - | - | - |
| Great-crested Flycatcher | 7 | - | 0.00 | 0 | 0.00 | - | - | - | - | - | - | 0 | - | - | - | - |
| Hairy Woodpecker | 11 | 2.76 | 0.00 | 0 | 0.00 | - | - | - | - | - | - | 0 | - | - | - | - |
| Least Flycatcher | 6 | 2.34 | 0.00 | 0 | 0.00 | - | - | - | - | - | - | 0 | - | - | - | - |
| Purple Finch | 1 | 0.12 | 0.00 | 0 | 0.00 | - | - | - | - | - | - | 0 | - | - | - | - |
| Red-breasted nuthatch | 2 | - | 0.00 | 0 | 0.00 | - | - | - | - | - | - | 0 | - | - | - | - |
| Ruby-throated Hummingbird | 1 | 1.61 | 0.00 | 0 | 0.00 | - | - | - | - | - | - | 0 | - | - | - | - |
| Swamp Sparrow | 2 | 0.03 | 0.00 | 0 | 0.00 | - | - | - | - | - | - | 0 | - | - | - | - |
| Swainson's Thrush | 1 | - | 0.00 | 0 | 0.00 | - | - | - | - | - | - | 0 | - | - | - | - |
| Tennessee Warbler | 1 | - | 0.00 | 0 | 0.00 | - | - | - | - | - | - | 0 | - | - | - | - |
| Traill's flycatcher | 6 | - | 0.00 | 0 | 0.00 | - | - | - | - | - | - | 0 | - | - | - | - |
| Yellow-bellied Flycatcher | 4 | - | 0.00 | 0 | 0.00 | - | - | - | - | - | - | 0 | - | - | - | - |
| Yellow-bellied Sapsucker | 18 | 2.41 | 0.00 | 0 | 0.00 | - | - | - | - | - | - | 0 | - | - | - | - |

*(Continued)*

**Table 2.** (Continued)

| | Birds | | | | | | Feeding larvae | | | | | | Feeding nymphs | | | | |
| Species | No. captured | Density (ind/ha) | Infestation prevalence | No. ticks collected | Infestation intensity[1] | Mean infectivity[2] | Pos. Bb[3] (n) | MLE IR pools[4] | No. ticks in pools | Pos. Bm[5] (n) | Pos. Ap[6] (n) | No. ticks collected | Tested (n) | Pos. Bb (n) | Pos. Bm (n) | Pos. Ap (n) |
|---|---|---|---|---|---|---|---|---|---|---|---|---|---|---|---|---|
| Yellow Warbler | 13 | 0.91 | 0.00 | 0 | 0.00 | - | - | - | - | - | - | 0 | - | - | - | - |
| Northern Flicker | 4 | - | 0.00 | 0 | 0.00 | - | - | - | - | - | - | 0 | - | - | - | - |
| **Subtotal** | **153** | **23.52** | **0.00** | **0** | **0.00** | - | - | - | - | - | - | **0** | - | - | - | - |

[1] Mean number of ticks per infested host in our sample

[2] Weighted mean between results from larvae tested individually and pooled unfed larvae, according to the number of ticks

[3] Bb: Number of *Borrelia burgdorferi* positive ticks

[4] Maximum-Likelihood Estimation Infection Ratio calculated from pools of unfed larvae

[5] Bm: Number of *Borrelia miyamotoi* positive ticks

[6] Ap: Number of *Anaplasma phagocytophilum* positive ticks.

the study. The prevalence of infection was lower for other pathogens, with values of 0.77% for *B. miyamotoi* and 2.26% for *A. phagocytophilum*. We did not find any co-infections in the questing nymphs.

We tested all 2,257 *I. scapularis* feeding larvae (including 1,226 ticks tested individually and 1,031 unfed ticks tested in pools), which were collected from 19 bird species and three small mammal species. *Borrelia burgdorferi* prevalence in feeding larvae (26.97%) was higher than in questing nymphs (9.10%, $\chi^2$ = 204.19, p < 0.001, Table 3). Similarly, the prevalence of *B. miyamotoi* was higher in feeding larvae (1.47%) than in questing nymphs (0.77%, $\chi^2$ = 3.98, p = 0.05). Larvae infected with *B. miyamotoi* were collected from three birds (one Hermith Thrush (*Catharus guttatus*), one Song Sparrow and one Veery (*Catharus fuscescens*); prevalence: 0.77%; Tables 2 and 3), 15 mice (prevalence: 1.47%, Table 3) and one chipmunk (prevalence: 13.64%, Table 3). *Anaplasma phagocytophilum* was also present at low levels in our samples, but more prevalent in questing nymphs (2.26%) than in feeding larvae (0.64%, $\chi^2$ = 29.04, p < 0.001, Table 3). Co-infections of *B. burgdorferi* and *B. miyamotoi*, were detected in 15/929 larvae tested individually and 4/197 pools (prevalence: 1.04%) feeding on seven white-footed mice and from 2/16 larvae tested individually (prevalence: 9.09%) collected from one chipmunk.

The prevalence of *B. burgdorferi* in larvae collected from 138 mice and 2 chipmunks was 29.73% and 27.27%, respectively (Table 3). None of the three larvae collected from shrews

**Table 3. Prevalence (%) of *B. burgdorferi*, *B. miyamotoi* and *A. phagocytophilum* in questing nymphs and larvae removed from hosts with 95% confidence intervals of exact binomial tests.**

| Pathogen | Questing nymphs | Feeding larvae | | | | | | | | |
|---|---|---|---|---|---|---|---|---|---|---|
| | | Mice | | | Chipmunks | | | Birds all sp. | | |
| | | Tested individually | Tested in pools | Combined | Tested individually | Tested in pools | Combined | Tested individually | Tested in pools | Combined |
| *B. burgdorferi* (*Bb*) | 9.1 | 46.93 | 12.22 | 29.73 | 37.50 | 0.00 | 27.27 | 17.63 | 5.67 | 14.19 |
| | [7.9–10.3] | [43.68–50.20] | [9.88–14.87] | [26.93–32.69] | [15.20–64.57] | [0.00–27.39] | [11.05–54.43] | [13.33–22.62] | [2.04–12.05] | [10.09–19.59] |
| | (201) | (436) | (85) | (521) | (6) | (0) | (6) | (49) | (5) | (54) |
| *B. miyamotoi* (*Bm*) | 0.8 | 2.37 | 0.57 | 1.47 | 18.75 | 0.00 | 13.64 | 1.08 | 0.00 | 0.77 |
| | [0.4–1.2] | [1.47–3.56] | [0.20–1.22] | [0.85–2.40] | [4.05–45.65] | [0.00–27.39] | [2.94–40.67] | [0.22–3.12] | [0.00–1.70] | [0.16–2.71] |
| | (17) | (22) | (5) | (27) | (3) | (0) | (3) | (3) | (0) | (3) |
| *A. phagocytophilum* (*Ap*) | 2.3 | 0.11 | 0.00 | 0.05 | 0.00 | 0.00 | 0.00 | 0.00 | 0.00 | 0.00 |
| | [1.7–3.0] | [0.00–0.60] | [0.00–0.21] | [0.00–0.41] | [0.00–0.00] | [0.00–27.39] | [0.00–7.47] | [0.00–0.00] | [0.00–1.70] | [0.00–0.49] |
| | (50) | (1) | (0) | (1) | (0) | (0) | (0) | (0) | (0) | (0) |
| Coinfections *Bb—Bm* | 0.00 | 1.61 | 0.45 | 1.04 | 12.50 | 0.00 | 9.09 | 0.00 | 0.00 | 0.00 |
| | [0.0–0.2] | [0.91–2.65] | [0.14–1.04] | [0.53–1.85] | [1.55–38.35] | [0.00–27.39] | [1.13–35.36] | [0.00–0.00] | [0.00–1.70] | [0.00–0.49] |
| | (0) | (15) | (4) | (19) | (2) | (0) | (2) | (0) | (0) | (0) |
| Coinfections *Bb—Ap* | 0.00 | 0.00 | 0.00 | 0.00 | 0 | 0.00 | 0.00 | 0.00 | 0.00 | 0.00 |
| | [0.0–0.2] | [0.00–0.00] | [0.00–0.21] | [0.00–0.10] | [0.00–0.00] | [0.00–27.39] | [0.00–7.47] | [0.00–0.00] | [0.00–1.70] | [0.00–0.49] |
| | (0) | (0) | (0) | (0) | (0) | (0) | (0) | (0) | (0) | (0) |
| **Number of ticks** | **2210** | **929** | **197** | **1126** | **16** | **1** | **17** | **278** | **24** | **302** |

Ticks partially fed to fully engorged with hosts's blood were tested individually, and unfed ticks were tested in pools per host. The numbers in brackets indicate confidence intervals and the numbers in parenthesis indicate number of positive ticks among those tested, for each category.

were infected with pathogens. The prevalence of *B. burgdorferi*-infected larvae collected from birds was 14.19% (Table 3), ranging from 0 to 100% (1/1 infected larva collected from a Dark-eyed Junco (*Junco hyemalis*)), depending on the species.

Of the three *I. muris* nymphs collected from birds, two were infected with *B. burgdorferi* (prevalence: 66.67%) and the remaining nymph was infected with *A. phagocytophilum* (prevalence: 33.33%).

## Hosts risk factors for infestation with blacklegged ticks and infectivity with *B. burgdorferi*

We found a positive relationship between the density of larvae collected by drag sampling and tick infestation density on hosts (Table 4). In birds, open-habitat species carried 2.56 times more [95% CI: 1.45–4.50] larvae than forest species. Ground-nesting species carried 2.86 times

**Table 4. Fixed effect parameter estimates for the best generalized mixed models of larval abundance on hosts (models 1 and 2) and the probability of transmission of *B. burgdorferi* from host to feeding larva (models 3 and 4) in 2017 and 2018**[*].

| Parameters | β | SE | P |
|---|---|---|---|
| **Model 1: Birds infestation** | | | |
| (Intercept) | -2.472 | 0.379 | <0.001 |
| Habitat | | | |
| *Open fields (vs. Forest)* | 1.052 | 0.270 | <0.001 |
| Nest location | | | |
| *Low (vs. High)* | 0.940 | 0.288 | <0.001 |
| Feeding behavior | | | |
| *Bark forager (vs. Ground forager)* | -1.117 | 0.591 | 0.059 |
| *In flight (vs. Ground forager)* | -0.192 | 0.006 | 0.997 |
| *Foliage gleaner (vs. Ground forager)* | -2.046 | 0.425 | <0.001 |
| Predicted density of questing larvae | 0.063 | 0.010 | <0.001 |
| **Model 2: Mice infestation** | | | |
| (Intercept) | -1.123 | 0.221 | <0.001 |
| Sex | | | |
| *Male (vs. Female)* | 0.413 | 0.135 | 0.002 |
| Predicted density of questing larvae | 0.123 | 0.010 | <0.001 |
| **Model 3: Birds infectivity** | | | |
| (Intercept) | -3.472 | 1.068 | 0.001 |
| Sex | | | |
| *Female (vs. Male)* | 1.382 | 0.505 | 0.006 |
| Habitat | | | |
| *Open fields (vs. Forest)* | 2.344 | 1.073 | 0.029 |
| **Model 4: Mice infectivity** | | | |
| (Intercept) | -1.006 | 0.228 | <0.001 |
| Sex | 0.427 | 0.226 | 0.059 |
| *Female (vs. Male)* | | | |
| Age | 1.299 | 0.214 | <0.001 |
| *Adults (vs. Juvenile)* | | | |
| Number of engorged larvae | 0.233 | 0.079 | 0.003 |

[*]The first models (1 and 2) were fitted with a negative binomial distribution, and the second (3 and 4) with binomial and beta binomial distributions respectively. For all models, the capture site was included as a random effect.

more [95% CI: 1.69–4.86] larvae than species that nest in trees or shrubs (Table 1). Feeding behavior was also explanatory in some cases; ground foragers carried 7.74 times more larvae [95% CI: 3.36–17.81] than foliage gleaners, but the differences were not significant when comparing other feeding behaviors with each other (multiple comparisons with Tukey adjustment, p = 0.05). For mice, sex of the host was the only intrinsic factor to be significantly associated with the number of ticks infesting hosts, where males carried on average 1.51 times more larvae [95% CI: 1.16–1.97] than females. In both cases, the data followed a negative binomial distribution and both models showed no significant residual spatial autocorrelation (bird model: p = 0.52, mouse model: p = 0.26).

The factors determining the probability of transmission of *B. burgdorferi* to larvae were different in birds and mice (Table 4). Sex of the host influenced transmission in both cases, but conversely: females were associated with a higher probability of infection in birds (OR: 3.98 [95% CI: 1.48–10.72]), whereas in mice, males were more likely to transmit infection (marginally significant association, p = 0.059, OR: 1.26 [95% CI: 1.09–1.93]) than females. Age affected this probability only in mice, where adults were at higher risk of transmitting infection than juveniles (OR: 3.67 [95% CI: 2.41–5.58]). As with the tick abundance, birds living in open-habitats were more likely than forest species (OR: 10.42 [95% CI: 1.27–85.38]) to infect feeding ticks. Finally, ticks that fed on a host for longer were more likely to transmit infection but this relationship was only significant for mice. Because of overdispersion in the mouse data (dispersion parameter = 2.09), we fitted the model using a beta binomial distribution [71]. For the bird model, the dispersion was less pronounced (dispersion parameter = 1.86) and the beta-binomial distribution did not improve the fit, so the binomial distribution was retained. Neither model showed significant residual spatial autocorrelation (bird model: p = 0.97, mouse model: p = 0.11).

## Contribution of hosts to *B. burgdorferi* transmission cycles

We compared *B. burgdorferi* reservoir potential of species for which we found feeding larvae infected by this pathogen, and for which we could obtain population density estimates in our study area. This analysis therefore excluded the Eastern chipmunks, for which we found infected larvae but captured insufficient animals to accurately estimate density.

We found infected larvae on five species of birds: Dark-eyed Junco, Song Sparrow, Chipping Sparrow (*Spizella passerina*), House Wren (*Troglodytes aedon*) and Hermit Thrush. The density of these species ranged from 0.27 to 1.57 individuals per hectare and the most abundant species in our study area were the Song Sparrow (n = 60) and Hermit Thrush (n = 36). The average abundance of larvae on birds of these five species was 1.47 larvae and ranged from 0.88 to 5.00 larvae, depending on the species (Table 5). The average infectivity of individuals ranged from 11 to 100% (the only larva found on Dark-eyed Juncos was infected) and from 19 to 20% by family. Comparing the species in which we detected *B. burgdorferi* by family, these represented between 1 and 11% of the relative reservoir potential, together totaling 18% of the estimated transmissions in this host community (Table 5).

A total of 450 individual white-footed mice were included in the analysis of reservoir potential and the estimated density of this species was 6.02 individuals per hectare. The average number of ticks on mice was 2.71 larvae per individual, the observed infectivity in mice was 30% and their reservoir potential, relative to the bird species was 82% of estimated transmissions (Table 5).

## Discussion

In this fine-scale field study, we investigated and compared the role of breeding birds to rodents in local transmission dynamics of *B. burgdorferi* s.s., *A. phagocytophilum* and *B.*

**Table 5. A comparison of the reservoir potential for the hosts individuals in which we detected *B. burgdorferi*, grouped by species and by family.**

| Species / Family[1] | Infestation | | | Infectivity | | Density | Reservoir potential (RP) | | Relative reservoir potential (RRP) | |
|---|---|---|---|---|---|---|---|---|---|---|
| | No. individuals examined[2] | Observed larvae density | Model-adjusted[3] | Average individuals infectivity[4] | No. feeding larvae analysed | Species density (ind/ ha) | RP-raw | RP-adjusted | RRP-raw | RRP-adjusted |
| *Passerellidae* | **78** | **1.19** | **0.61** | **0.20** | **201** | **2.00** | **0.47** | **0.24** | **0.08** | **0.06** |
| Chipping Sparrow | 16 | 0.88 | 0.70 | 0.11 | 19 | 0.57 | 0.05 | 0.04 | 0.01 | 0.01 |
| Dark-eyed Junco | 2 | 0.50 | 0.54 | 1.00 | 1 | 0.27 | 0.14 | 0.15 | 0.02 | 0.03 |
| Song Sparrow | 60 | 1.30 | 0.59 | 0.20 | 181 | 1.16 | 0.31 | 0.14 | 0.05 | 0.03 |
| *Turdidae* | **36** | **1.86** | **1.62** | **0.19** | **75** | **1.57** | **0.55** | **0.48** | **0.09** | **0.11** |
| Hermit Thrush | 36 | 1.86 | 1.62 | 0.19 | 75 | 1.57 | 0.55 | 0.48 | 0.09 | 0.11 |
| *Troglodytidae* | **2** | **5.00** | **1.64** | **0.20** | **10** | **0.19** | **0.19** | **0.06** | **0.03** | **0.01** |
| House Wren | 2 | 5.00 | 1.64 | 0.20 | 10 | 0.19 | 0.19 | 0.06 | 0.03 | 0.01 |
| *Cricetidae* | **450** | **2.71** | **1.99** | **0.30** | **1842** | **6.02** | **4.87** | **3.58** | **0.80** | **0.82** |
| White-footed mouse | 450 | 2.71 | 1.99 | 0.30 | 1842 | 6.02 | 4.87 | 3.58 | 0.80 | 0.80 |

[1]Data are aggregated by species or family. The relative reservoir potential (RRP) indices sum up to 1 in both cases.

[2]Excludes recaptured individuals and those for whom there was missing data in the variables included in the infestation model.

[3] Rates adjusted according to predictions of the number of feeding larvae for Julian day 200 of each sampling year, based on negative binomial GLMMs models of host infestation.

[4]Weighted means of results from larvae tested individually and pools of unfed larvae.

*miyamotoi*, which are emerging pathogens in southeastern Canada. We provided a first record of *B. miyamotoi* detected from larvae collected from birds. Our intensive sampling of breeding birds allowed us to highlight the contribution of this host group to the transmission cycle of the most prevalent tick-borne pathogen, *B. burgdorferi*. Nearly one third of captured birds were infested with *I. scapularis*, five species could efficiently transmit *B. burgdorferi* to larvae during their blood meal and based on these data we estimated that birds may account for approximately one fifth of the infected host-seeking *I. scapularis* nymphs in the park. In small mammals, white-footed mice dominated the sample and accounted for the remaining 80% of estimated nymph infections by *B. burgdorferi*. Given that this tick stage is associated with a high risk of disease transmission to humans, this study highlights the importance of acquiring more knowledge about alternative reservoir hosts, the strains they can transmit, and the possible impacts on the ecology and epidemiology of tick-borne diseases.

## Pathogen prevalence in *I. scapularis* ticks

We detected *B. burgdorferi*, *B. miyamotoi* and *A. phagocytophilum* in host-seeking *I. scapularis* nymphs, signifying a possible risk of infection for human populations frequenting the park where the study took place. The most prevalent pathogen was *B. burgdorferi*, followed by *A. phagocytophilum* and *B. miyamotoi*, which had prevalence in unfed nymphs comparable to rates reported elsewhere in Canada [72, 73].

In feeding *I. scapularis* larvae, *B. burgdorferi* was also the dominant pathogen, followed by *B. miyamotoi*, and was observed in ticks collected from chipmunks, mice, and birds (a Hermit Thrush, a Song Sparrow, and a Veery). Another North American study also detected *B. miyamotoi* from ticks (*I. dentatus*) collected from passerines [74]. Since *B. miyamotoi* can be

transmitted transovarially [33] we cannot infer the reservoir competence of chipmunks and birds from these results. A study performed on a sample of hunter-harvested white-tailed deer from Wisconsin, USA suggested that deer may also be a reservoir for *B. miyamotoi* [75]. Future studies considering all these different species, as well as xenodiagnostic experiments [28], will be necessary to better characterise the reservoirs of this pathogen. Although the prevalence of this pathogen was lower than those reported in endemic areas of the northeastern USA [28, 33, 74], the fact that it was detected in resident hosts in addition to questing nymphs suggests that it may have begun to circulate locally in this area. In order to establish viable populations into new areas, tick-borne pathogens depend on the local host community to efficiently reproduce [76]. Thus the detection of an emerging pathogen only from questing ticks, as has been the case to date in other published studies in Canada [1, 72, 73], may be attributable to founder events (e.g. dissemination of immature infected ticks by migratory birds [77]) that do not necessarily demonstrate the establishment of a local transmission cycle.

*Anaplasma phagocytophilum* was present in larvae collected from hosts in lower proportion than in questing nymphs, with only one tick infected with *A. phagocytophilum* collected from a white-footed mouse. Although our analyses did not attempt to distinguish the Ap-ha strain from the Ap-variant 1 strain, the detection of an infected larva from a mouse suggests the Ap-ha variant, given the demonstrated inability of mice to transmit the Ap-variant 1 [78]. The very low proportion of infected larvae collected from hosts suggests that other hosts not sampled in our study may act as reservoirs for this pathogen, notably white-tailed deer which are associated with the maintenance of Ap-variant 1 [79].

## Infestation with blacklegged ticks and *B. burgdorferi* prevalence in ticks collected from birds

Nearly 30% of the birds captured in our study (belonging to 28 species) were infested with *I. scapularis* ticks. Among these, we identified five bird species that can effectively transmit *B. burgdorferi* to naïve larvae, with infectivity rates ranging from 11 to 20%. Comparing these rates by species, they are all at least two times higher than those reported in a recent meta-analysis by Loss et al. [22] on the role of birds in tick-borne pathogens dynamics in North America.

For the remaining 23 infested species, we could not demonstrate that these species were capable of transmitting *B. burgdorferi*. Of these, 14 were parasitized by both larvae and nymphs and for 5 of these, none of the ticks tested positive for *B. burgdorferi*. Since the prevalence of *B. burgdorferi* in questing nymphs is close to 10%, this discrepancy in prevalence in feeding nymphs among hosts could mean that certain species can clear tick infection during a blood meal, for example through activation of the complement pathway by the host's innate immune system [59, 80]. It had previously been suggested that certain bird species may act as zooprophylactic hosts [10, 16], but more studies are still needed to associate the patterns observed here and the immune mechanisms involved.

For the other nine species for which we could not demonstrate the ability to transmit *B. burgdorferi* via infection of feeding larvae, we did detect the pathogen from feeding nymphs, with prevalence of infection comparable to or higher than that observed in questing nymphs. This was the case for the Black-and-White Warbler (*Mniotilta varia*), Black-capped Chickadee (*Poecile atricapillus*), Common Yellowthroat (*Geothlypis trichas*), Veery and White-breasted Nuthatch (*Sitta carolinensis*), suggesting that these species may be ineffective as reservoirs, but are not zooprophylactic hosts. This may be due to low bacterial loads in the blood of the birds or the infectious periods being too short or variable to allow efficient transmission and enzootic maintenance of the pathogen [59]. While we could not find any mention of bird-to-larva

transmission of *B. burgdorferi* for Black-capped Chickadees and White-breasted Nuthatches in the literature, it has been reported that Black-and-White Warblers can infect a small proportion of larvae [22], whereas Common Yellowthroats and Veerys are considered efficient transmitters [18, 22]. The differences between our results and those from other parts of North America could be due to regional differences in *B. burgdorferi* strains, which may be more or less well-adapted to certain host species, as observed in Europe with *B. burgdorferi* s.l. [59]. It may also be that we were simply not able to collect enough individuals of these host species and enough larvae from them to allow us to reliably detect the infection in them.

## Hosts risk factors for infestation with blacklegged ticks and infectivity with *B. burgdorferi*

Birds associated with open habitats were more densely infested and more likely to infect larvae with *B. burgdorferi* than birds from forested habitats, which contrasts with results from other North American studies that reported higher numbers of ticks per bird when animals were collected in large and dense forest patches [15, 81]. However, these studies were conducted in different ecological settings than ours (in terms of climate, plant and wildlife species composition of forests and open habitats), therefore more studies will be needed to explain these divergent results. Ground foragers were more densely infested than other birds, while no such difference between groups was found regarding nest location. This is consistent with the results of several other studies [8, 22, 81–84] that found foraging behavior to be more important than time spent on the ground per se (e.g., on a nest) for explaining tick acquisition [15, 22]. The majority of bird species (80%, 4/5) and individuals (83%, 97/117) for which we could demonstrate reservoir competence were also ground foragers. The effect of this specific behavior on the chances of infecting larvae could not be highlighted by our model for individual bird infectivity (too few observations in the other behavior categories). However, we found high proportions of individuals and species belonging to this group among those in which pathogen transmission to larvae was detected. This suggests that this behavior may also be determinant in species' contribution to the transmission cycle of *B. burgdorferi*. Indeed, increasing the prevalence of the infested hosts and the intensity of their infestation by ticks generally leads to an increase in their chances of acquiring tick-borne pathogens and retransmitting them to feeding ticks [59].

Adult male mice were both more densely infested and more likely to transmit *B. burgdorferi* to larvae than female and juvenile mice, which is consistent with previous studies and the explanation that males contact ticks more often because they have a larger home range and generally more exploratory behavior than female and juvenile mice [35, 56]. In birds, sex of the animal was not a predictor of abundance of ticks, but the probability of infecting larvae was higher in female than in male birds. This may be due to a different energy balance between females and males (e.g. high energy cost of reproductive effort by females), which may in turn affect the strength of immune suppression mechanisms [85]. Age of birds did not influence infestation or infectivity rates, contrary to previous observations that juveniles were more densely infested [86] and transmitted infection more often than adults [16].

## Contribution of hosts to *B. burgdorferi* transmission

We estimated that approximately one fifth of infected nymphs in the park would have acquired their infection from a bird, assuming that the survival rate from larvae to nymphs is equivalent regardless of the species used as host. Excluding the Dark-eyed junco for which we obtained a single positive larva, the bird species with the greatest reservoir potential for *B. burgdorferi* was the House Wren, followed by the Hermit Thrush, Song Sparrow and Chipping Sparrow. These

are species for which previous studies had already suggested a reservoir competency [15, 18, 22], but whose contribution was considered low compared to other common reservoir bird species, such as American Robin and Northern Cardinal (*Cardinalis cardinalis*) [18, 22]. We estimated that white-footed mice are likely responsible for infecting most of the nymphs in the park, which is in line with current evidence that this species is the principal reservoir of *B. burgdorferi s.s.* in eastern North America [87]. This estimate of species reservoir potential was a function of three parameters: the average number of larvae infesting individual hosts, their ability to transmit *B. burgdorferi* to uninfected ticks, and their density in the habitat. On average, larval abundance on reservoir bird species was slightly lower (1.47 larvae/individual) than that of mice (2.71 larvae/individual). Nevertheless, these data show that birds can be frequently used as a host for immature stages of *I. scapularis*, contrary to what has been found in other studies [21]. Also, the level of infectivity of these reservoir bird species was about 10% lower than that of mice. Finally, the density of mice in the park was much higher than that of the five reservoir bird species. Low relative density therefore appears to be the most limiting factor in the contribution of birds to transmission dynamics, as suggested by other studies [18, 19]. Since the most important avian reservoirs were associated with open-field habitats and these covered only a small proportion of the study site, it would be interesting to repeat this study in different habitats and with different species assemblages to determine how bird's contribution to *B. burgdorferi* transmission cycles varies. Nevertheless, our results underline the fact that multiple host species contribute to the maintenance of *B. burgdorferi* transmission cycles in nature and thus alternative hosts will be important to consider to ensure optimal effectiveness of control efforts targeted to wildlife reservoirs, such as acaricide treatments or vaccines [23–25]. Furthermore, genetic diversity of *B. burgdorferi* maintained by these alternative hosts [18, 87] will also have to be investigated, since different strains may cause clinically or diagnostically different disease outcomes for humans [88].

This observational field study has some limitations. This portrait of *B. burgdorferi* transmission patterns is limited to hosts targeted by our sampling design whereas other species, although common in the park and with reservoir capabilities previously shown, were excluded because of their scarcity in our samples. This was the case for birds like American Robins, Blue Jays (*Cyanocitta cristata*), Common Yellowthroats and Northern Cardinals [18, 22] and for small mammals like Eastern Gray Squirrel (*Sciurus carolinensis*) and Eastern chipmunk. Also, the duration of host infectiousness could not be accounted for in our study. However, results from laboratory assays suggest temporal variability in the infectiousness of birds for larval ticks [14]. While the long period of infectivity of mice is well documented in some studies [59], more research is needed to explore how this varies in different bird species in natural settings. Finally, the temporality in host availability was not considered. It has been documented that mouse populations exhibit interannual fluctuations [89, 90] and shifts in the seasonality of abundance peaks [91]. Thus birds, whose population sizes and seasonal activity patterns are more stable from year to year, could play a stabilizing role in the *B. burgdorferi* enzootic cycle in years when mouse populations are at their lowest levels [16] or asynchronous with the seasonality of questing ticks [91]. Finally, we did not detect any pathogens in biopsies taken from birds, although many larvae feeding upon birds were infected with *B. burgdorferi* or *B. miyamotoi*. Because *B. burgdorferi* is not capable of transovarial transmission in *I. scapularis* ticks [33], infections detected in feeding larvae must have been acquired from the host. There may have been undefined inhibitors in the skin of the birds that somehow inhibited the efficacy of the PCR on skin relative to attached ticks. However it is more likely that the location that the biopsy was taken from influenced the results. For example, we chose this biopsy site because it is minimally invasive when handling animals. However, taking biopsies near the site where ticks attach on the animal but cannot be groomed off (on the head of birds) would be

preferable in future studies. It is thought that in infected hosts, *B. burgdorferi* is attracted to the tick feeding site by tick salivary proteins, and multiplies in the tick feeding lesion (reviewed in [92]). Our findings may be consistent with this as no ticks were found near the biopsy sites we chose, while, biopsies from the heads of birds have been used successfully to detect some genospecies of *B. burgdorferi* s.l. in Portugal [93]. The processes of dissemination of tick-borne pathogens in bird tissues should be further investigated.

## Conclusion

This study illustrates the transmission of *B. burgdorferi*, *A. phagocytophilum* and *B. miyamotoi* within a host community typical of the ecological context in which the geographic range expansion of their vector, *I. scapularis*, is currently occurring in northeastern North America. It is the first empirically based quantitative assessment of the contribution of mice and breeding birds as reservoirs of the Lyme disease pathogen in North America. Our results support the relevance of considering the role of hosts other than the white-footed mouse in eco-epidemiological studies of tick-borne diseases. The next steps will be to continue acquiring knowledge on the diversity of reservoir hosts of these emerging pathogens across different locations in North America, and to investigate their genetic diversity and potential strain-host associations. This information will be essential for improving management of the risk associated with emerging tick-borne diseases.

## Supporting information

**S1 File. Original datasets used in the study.**
(XLSX)

## Acknowledgments

We would like to thank the employees of the Mont-Saint-Bruno National Park for the logistic support during the sampling and all those who participated in the field efforts. Also a special mention to Dominique Dufault for the support with the bird captures.

## Author Contributions

**Conceptualization:** Ariane Dumas, Catherine Bouchard, Pierre Drapeau, Nicholas H. Ogden, Patrick A. Leighton.

**Data curation:** Ariane Dumas.

**Formal analysis:** Ariane Dumas.

**Funding acquisition:** Nicholas H. Ogden, Patrick A. Leighton.

**Investigation:** Ariane Dumas, Antonia Dibernardo, L. Robbin Lindsay.

**Methodology:** Ariane Dumas, Antonia Dibernardo, L. Robbin Lindsay.

**Project administration:** Ariane Dumas, Patrick A. Leighton.

**Resources:** L. Robbin Lindsay, Nicholas H. Ogden, Patrick A. Leighton.

**Supervision:** Nicholas H. Ogden, Patrick A. Leighton.

**Validation:** Nicholas H. Ogden, Patrick A. Leighton.

**Visualization:** Ariane Dumas.

**Writing – original draft:** Ariane Dumas.

**Writing – review & editing:** Catherine Bouchard, Antonia Dibernardo, Pierre Drapeau, Nicholas H. Ogden, Patrick A. Leighton.

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
