## [Decision Letter · Decision Letter 0]

20 Dec 2021

PONE-D-21-34840Transmission patterns of tick-borne pathogens among birds and rodents in a forested park in southeastern CanadaPLOS ONE

Dear Dr. Dumas,

Thank you for submitting your manuscript to PLOS ONE. After careful consideration, we feel that it has merit but does not fully meet PLOS ONE’s publication criteria as it currently stands. Therefore, we invite you to submit a revised version of the manuscript that addresses each of the points raised during the review process.

We look forward to receiving your revised manuscript.

Kind regards,

Brian Stevenson, Ph.D.

Academic Editor

PLOS ONE

Journal Requirements:

2.We note that you have stated that you will provide repository information for your data at acceptance. Should your manuscript be accepted for publication, we will hold it until you provide the relevant accession numbers or DOIs necessary to access your data. If you wish to make changes to your Data Availability statement, please describe these changes in your cover letter and we will update your Data Availability statement to reflect the information you provide.

a) You may seek permission from the original copyright holder of Figure 1 to publish the content specifically under the CC BY 4.0 license.  

Natural Earth (public domain): http://www.naturalearthdata.com/.

Reviewers' comments:

Reviewer's Responses to Questions

**Comments to the Author**

1. Is the manuscript technically sound, and do the data support the conclusions?

Reviewer #1: Yes

Reviewer #2: Yes

Reviewer #3: Yes

2. Has the statistical analysis been performed appropriately and rigorously? 

Reviewer #1: Yes

Reviewer #2: Yes

Reviewer #3: Yes

3. Have the authors made all data underlying the findings in their manuscript fully available?

Reviewer #1: Yes

Reviewer #2: Yes

Reviewer #3: Yes

4. Is the manuscript presented in an intelligible fashion and written in standard English?

Reviewer #1: Yes

Reviewer #2: Yes

Reviewer #3: Yes

5. Review Comments to the Author

Reviewer #1: In this manuscript, the authors investigate the abundance of Borrelia burgdorferi, Borrelia miyamotoi, and Anaplasma phagocytophilum in avian and rodent species in Mont Saint-Bruno National Park to compare the contribution of each species to pathogen transmission as reservoir species and to identify factors contributing to tick infestation and pathogen infectivity in the surveyed species. The authors conclude that while rodents mostly contribute to the reservoir species in the park, birds also serve as reservoir species providing importance for further study of birds as reservoirs for B. burgdorferi and B. miyamotoi. The authors also identify risk factors in avian and rodent populations (sex, habitat, foraging behavior) that may contribute to tick infestation and B. burgdorferi infectivity. I have the following relatively minor critiques concerning the manuscript:

The second paragraph (lines 60-64) is underdeveloped and would be improved with more detail. I think it would be nice to elaborate on other rodents identified as reservoir species, examples of susceptibility of infection, infectious period, and/or ubiquity in ecosystems that are mentioned in this paragraph. As it stands, this paragraph is only two sentences.

Figure 1 may be improved if it could be presented in higher resolution. My full-size print of the figure is very blurry, but I can see the sampling sites well enough.

In the “Diagnostic testing for tick-borne pathogens” section (starting at line 170) there is no reference or sequences provided for primers used for the real-time PCR assay used to screen for pathogens. These may be useful to other groups looking to study pathogen prevalence. Furthermore, there is no mention of where biopsies were sampled on birds and mice. For instance, it would be interesting to know if the bird biopsies taken from birds with infected larvae (mentioned lines 336-337) were taken near the tick feeding site or distal to the site. This could have some implication on potential pathogen dissemination defects in birds.

In lines 341-342, I think the authors meant to say “the numbers in parenthesis indicate number of ticks tested” instead of brackets. It is not specifically mentioned what the numbers in brackets are, but I assume this is the range in percentage?

Reviewer #2: Comments also included as a separate document.

Major Comments

Dumas et al. describe the role of small mammals and birds as tick hosts and tick-borne pathogen reservoirs in southeastern Canada. Most notably, they found that five bird species can contribute 18% of the estimated transmissions of Borrelia burgdorferi to Ixodes scapularis larvae feeding on the birds. In addition, they also estimated that the reservoir potential of white-footed mice (Peromyscus leucopus) was 82% of estimated transmissions.

The manuscript is well-written and shows a lot of promise to improve the field of tick-borne disease ecology, specifically in Canada. The paper could improve on clarifying their methods and statements overall, which I included in the Minor Comments.

Not necessarily a weakness in the study itself, but I was wondering if the authors had a reason for only trapping and mist netting during a specific time of the year. Presumably, these animals are pretty active throughout much of the year outside of the summer season. Would other tick species or life stages be found on these animals during the spring or fall, thus possibly contributing other pathogens at other time points of the year? Are the models and statistics held true outside of the months where trapping and mist netting occurred? I saw that the authors conducted tick dragging May-October, but tick dragging does not seem to result in a high diversity of tick species compared to ticks found on animals and is not a replacement for trapping or mist-netting animals to find ticks.

Interestingly, the authors did not detect any pathogens in the bird biopsies, but found that the larval ticks were infected with pathogens. While the Discussion noted that the birds can possibly clear the infection, this still does not explain how the ticks were infected with the pathogen if the bird is cleared of infection by the time it was sampled. Larval ticks only feed for a few days, so is it possible that birds are clearing pathogens before the larvae are replete? Do the authors have data showing that more fully-engorged ticks have a higher pathogen load compared to a tick that just started feeding? This could provide strong evidence for their theory, where as a host is clearing the infection, ticks that fed sooner would be expected to have a higher pathogen load compared to those that started feeding later.

Furthermore, I do notice that more emphasis is given to the data on birds, while white-footed mice and other small mammals do not seem to garner the same level of attention. While I do agree that birds have been understudied as a tick host/tick-borne pathogen reservoir, I still think that the small mammal component is informative. If your emphasis is on birds, then the small mammal component should be removed to make the paper more focused and succinct. I personally think that both should be kept in the paper, but they should also be equally represented.

Regardless, I commend the authors’ efforts for their long-term field work and for creating sophisticated models to determine host risk factors for tick infestation and pathogen infectivity as well as the reservoir potential. I certainly learned something new and I know that this work will be important as we continue to progress our knowledge of hosts for ticks and reservoirs of tick-borne pathogens.

Minor Comments

• Line 33: Add sample sizes into the abstract.

• Line 36: There doesn’t seem to be any information on the small mammal infestation and infection data or the reservoir potential value in the abstract. Was there a reason to exclude it or could you add it to the abstract to make it more representative of your data?

• Lines 54-55: Diseases should be lowercased..

• Lines 54-55: The last part of this statement (“…caused by Borrelia miyamotoi”) makes it sound like all of the diseases you listed are caused by B. miyamotoi. Maybe instead you could list the relevant pathogens that cause the disease in parentheses after the disease is mentioned.

• Line 88 (and elsewhere): I think common bird names should be lowercased unless they are named after someone or a location. In addition, you might want to consider putting the scientific names of birds in parentheses after the common name is listed the first time. This was done for the small mammals in the Introduction and Results sections, so I think it should also be done for the birds for consistency.

• Lines 127-130: What were the methods for tick dragging and storing the ticks before pathogen testing? Are they included in another reference? If so, the reference should be added. Figure 1 is also listed in the paragraph, but I don’t see the caption for it anywhere in the review packet.

• Lines 132-143: I noticed that the Ethics Statement of the review packet included more information about trapping and bird mist netting and this information should be added to the manuscript itself as well.

• Lines 141-143: Do you think DNA degradation of your sample is possible if the sample is kept in ethanol and at room temperature for long periods of time?

• Line 153: When luring the birds to the nets, do you think you might have biased your mist netting results to specific species? Do you suspect that using sound lures would decrease capture success of other bird species that were not used as playback?

• Lines 173-174: “… and a subset of up to 30 nymphs per infested host species for nymphs collected from birds.” What about the samples from mice?

• Line 175: “Unfed larvae from individual hosts…” Do you mean that the larvae were pooled all together for pathogen testing? Or do you mean that you unfed larvae from the same individual/host were pooled together?

• Lines 180-182: Gene names should be italicized (23S, msp2, ospA/flab). “spp.” should not be italicized. Add a comma before “respectively.”

• Lines 182-184: The way the sentence is structured currently, it sounds like you tested samples from 2016 even though the study took place in 2017 and 2018. I would rephrase it so that you say you didn’t test any samples in 2018 since you did not find any positive samples in your first year of study in 2017. That being said, was there a reason why you didn’t test samples from both years? Wouldn’t results from 2017 be more or less independent of what you might find in 2018? In other words, what you find in 2017 is not necessarily representative of what you find in 2018.

• Lines 188-194: Instead of Chi-square analyses, would it be more accurate to test for significant differences using a model, especially since you are testing various origins. The origins could be your covariates and the presence/absence of the pathogen could be the outcome of the model.

• Lines 230-231: If you used any R packages for your statistical analyses, I recommend adding and referencing those packages in the manuscript.

• Lines 230: The sentence structure makes it sound like the DHARMa package conducts Moran’s I tests, but I don’t think it’s used for that. In this case, I would break up the sentence into two sentences and for the former, list the package you used for Moran’s I (see previous comment). R packages should also be italicized.

• Lines 253-254: Is it possible to assume that tick attachment between recaptures could happen, even within 24 hours? If all ticks were removed and you found ticks the next day, would this event be considered independent and could thusly be included in your analyses?

• Line 261: “The white-footed mouse…”

• Line 317: “Ticks were found on 245 (28.86%) birds, with the majority of these ticks…”

• Table 2: I may have missed this in the manuscript, but why are some of the densities not calculated for some birds? They either have a blank or a dash in them, but multiple birds were captured. Would it also be possible to have a similar table for the small mammal captures to succinctly describe those results as well?

• Line 341: Similar to a previous comment I made, were unfed ticks tested in pools per host? Or were all unfed ticks pooled together? I’m assuming the former, but this should be clarified here and in the main text of the manuscript.

• Lines 341-342: I’m not sure if this is PLOS One’s guidelines, but should this information go with the title as a caption? I feel like I would have understood the table better had I known this information before reading the table.

• Lines 344-346: The sentence sounds a little awkward towards the end, so I would suggest something like the following: We tested 2210 questing nymphs (63.52% of the total questing nymphs collected) and found an overall prevalence of 9.10% for B. burgdorferi across the three years of the study.

• Line 349: “We tested all 2257 I. scapularis feeding larvae…”

• Lines 370-371: What did open-habitat bird species carry more of? Ticks overall? Or certain life stages of ticks?

• Lines 370-372: The sentence is a little confusing, but based on my interpretation of the sentence, I think the sentence should be broken up into two separate sentences. One sentence compares open-habitat bird species and forest species, and the second sentence describes ground-nesting species and species that nest in trees and shrubs.

• Lines 376-378: “For mice, sex of the host was the only intrinsic factor to significantly affect the number of ticks infesting hosts, where males…”

• Table 4: Are some of the columns misaligned? Things like “Intercept,” “Predicted density of questing larvae,” and “Number of engorged larvae” look like they should be aligned with the rest of the table. I would also suggest moving the Infectivity Models to a separate table. When you reference them in the paper, it can get a little confusing as to which table you are referencing. I also almost missed the “Infectivity models” header when reading the table the first time. So either make two separate tables or make the header more noticeable.

• Line 405: “… but captured insufficient animals…”

• Table 5: I think it might look a little better if you somehow moved the family names and associated data up as a subheader and then list all of the bird species under that family. Then you have another subheader with the next family and list all those individuals under that family and so on and so forth.

• Lines 430-432: This was only briefly mentioned in the Results and B. miyamotoi results were combined for both small mammals and birds – I think finding the first report of B. miyamotoi in larvae from birds is super neat and should be emphasized in a separate sentence in the Results, if possible. That being said, to make the jump that B. miyamotoi circulates in bird populations at the study site might be a reach considering that none of the bird biopsies came up positive for the pathogen.

• Line 434: “most prevalent tick-borne pathogen, B. burgdorferi.”

• Lines 461-465: This sentence can probably be separated into two sentences.

• Lines 477-478: Again, can we really say this if the birds tested negative? Based on the explanation in lines 484-487, it sounds like birds can clear and infection during a blood meal, but like my previous comment mentioned, larval ticks only feed for a few days at a time. Are there references that note how quickly birds can clear infection and if this differs per species?

• Lines 504-506: I actually thought B. burgdorferi seemed pretty common in your ticks from birds, with infectivity rates between 11-20%. Even if the birds were not positive themselves, the ticks coming from the birds show that the pathogen is pretty prevalent.

• Lines 512-515: I think “and” is the wrong transition word. Maybe using “therefore” or “and thusly” might be better.

• Lines 520-525: This entire sentence is rather wordy and gets long-winded when trying to read it – could there be a way to shorten it or to break it up into a couple of sentences?

• Lines 534-536: Do you happen to have a reference supporting the claim about different energy balances between male and female mice and how it contributes to different immune suppression mechanisms?

• References: Some species/group names should be properly formatted (italicized, capitalized, etc.). See references 28, 35, and 64. Those were the ones I quickly found, but another review might be a good idea.

Reviewer #3: Dumas et al. conducted a study of tick-borne pathogens among birds and rodents in a forested park near Montreal, Quebec, Canada. This park is visited by many people and reports of B.burgdorferi transmission has been documented here. This study was conducted between 2016 and 2018 collecting questing ticks as well as ticks found on captured birds and rodents. Molecular analyses were conducted to determine if the ticks or vertebrate hosts were positive for Anaplasma phagocytophilum, Borrelia (Borreliella) burgdorferi, and Borrelia miyamotoi. The study was focused on determining the role of birds as potential reservoir hosts of these pathogens and estimated that within this park 5 bird species, identified as reservoir hosts, likely contributed to one-fifth of the B. burgdorferi infected nymphs with the remainder contributed by the white-footed mouse. This study identifies I. scapularis infestations in birds and indicates the ability of birds to contribute to the enzootic maintenance of B. burgdorferi in North America.

Overall, the study was well designed and executed. I only have a few clarification questions regarding methodology which may require further elaboration in the text.

Line 174-175 I assume engorgement was determined visually

Line 271-276 Were the point counts conducted both visually and aurally? How were the bird calls identified? Were recordings made and compared to sonograms or did someone identify the birds by ear? Were the point counts conducted at the same time of day each time or at different times?

The writing was good overall and I have only a few comments:

I noticed that occasionally there would be long, cumbersome sentences where the impact of the sentence would be enhanced by breaking it up into multiple sentences. For example lines 42-45, 107-112, 460-465, 591-595.

I am not sure if PLoS has an opinion on the capitalization of common names of birds since there is some debate about whether or not that is appropriate. However, please be consistent throughout the manuscript text and tables when capitalizing these names.

Lines 180 and 182 have gene names that should be italicized

For R packages, please indicate the version used if available

Line 261 "White-footed mouse" is capitalized but all other references in the text have

"white" as lower case

In table 2, please define what "infestation intensity" is.

Lines 341-342 The numbers that is referred to here is in parentheses whereas the confidence intervals are in brackets

Line 344 2210 needs a comma: 2,210

Line 349 2257 and 1226 also need commas: 2,257 and 1,226

Line 405 capture should be captured

Table 5 Under the 'Infestation' heading the 'No' needs a period 'No.'

Lines 446-450 It is not necessary to include this unpublished information and these lines could be removed.

Line 473 White-tailed deer may also be a reservoir for B. miyamotoi and might be worth mentioning here or considering for future studies. Han, S., Hickling, G. J., & Tsao, J. I. (2016). High prevalence of Borrelia miyamotoi among adult blacklegged ticks from white-tailed deer. Emerging infectious diseases, 22(2), 316.

The conclusions were sound and supported by the data but I do have one question concerning a statement made in lines 552-555. The authors state that there were twice the amount of larvae carried by birds compared to white-footed mice but I can't seem to find which data support that in the results.

6. PLOS authors have the option to publish the peer review history of their article (what does this mean?). If published, this will include your full peer review and any attached files.

Reviewer #1: No

Reviewer #2: No

Reviewer #3: No

---

## [Author Response · Author response to Decision Letter 0]

3 Feb 2022

Dear Dr Stevenson, 

Thank you for considering our manuscript. Please find below our response to the specific comments raised.

Comments by Academic editor: 

“If applicable, we recommend that you deposit your laboratory protocols in protocols.io to enhance the reproducibility of your results. “

Protocols will be available by contacting the laboratory directly. 

We revised the manuscript to ensure that it met PLOS ONE's style requirements.

We wish to make a change to the data availability statement: we meant that the data are accessible as supplementary material (S1 File), not through an online repository.

3. We note that Figure 1 in your submission contain [map/satellite] images which may be copyrighted. 

(…) supply a replacement figure that complies with the CC BY 4.0 license. Please check copyright information on all replacement figures and update the figure caption with source information. If applicable, please specify in the figure caption text when a figure is similar but not identical to the original image and is therefore for illustrative purposes only.

Caption corrected to mention the OpenStreetMap copyright (top left insert) and ecoforestry/landcover data (right and down left inserts). Original ecoforestry/landcover data available at https://www.foretouverte.gouv.qc.ca/ and the license is CC-BY 4.0. (see https://www.donneesquebec.ca/recherche/fr/dataset/resultats-d-inventaire-et-carte-ecoforestiere) 

4. Please review your reference list to ensure that it is complete and correct. 

We have revised the reference list.

Comments by Reviewer #1: 

1.1. The second paragraph (lines 60-64) is underdeveloped and would be improved with more detail. I think it would be nice to elaborate on other rodents identified as reservoir species, examples of susceptibility of infection, infectious period, and/or ubiquity in ecosystems that are mentioned in this paragraph. As it stands, this paragraph is only two sentences.

Modifications made.

1.2. Figure 1 may be improved if it could be presented in higher resolution. My full-size print of the figure is very blurry, but I can see the sampling sites well enough.

Resolution of the figure has been increased to 300 dpi, which now meets PlosOne requirements.

In the “Diagnostic testing for tick-borne pathogens” section (starting at line 170) there is no reference or sequences provided for primers used for the real-time PCR assay used to screen for pathogens. These may be useful to other groups looking to study pathogen prevalence. Furthermore, there is no mention of where biopsies were sampled on birds and mice. For instance, it would be interesting to know if the bird biopsies taken from birds with infected larvae (mentioned lines 336-337) were taken near the tick feeding site or distal to the site. This could have some implication on potential pathogen dissemination defects in birds.

We are confused by this comment as the primers and references for same were already provided in text as follows: ”Briefly, extracted DNA was screened using a duplex real-time PCR assay targeting the 23S and msp2 genes of Borrelia spp. and A. phagocytophilum respectively (Courtney et al., 2004). Borrelia-positive samples were subsequently tested for B. burgdorferi and B. miyamotoi using a second ospA/flab duplex assay (Tokarz et al., 2017).” Please let us know if we are missing something. 

However, we noticed while reviewing that we did not include the primers used for mice species identification. We thus added this information in the manuscript (Page 10, Line 216).

The information about the location of biopsy collection on birds (wing: prepatagial membrane) and mice (ear) can be found in Materials and Methods, section Field. We added the location of ticks on the body of the hosts in the Results section, and more explanations for our pathogen testing results in bird biopsies in the discussion. 

1.3. In lines 341-342, I think the authors meant to say “the numbers in parenthesis indicate number of ticks tested” instead of brackets. It is not specifically mentioned what the numbers in brackets are, but I assume this is the range in percentage?

Modification made.

Comments by Reviewer #2: 

Major Comments: 

2.1. Not necessarily a weakness in the study itself, but I was wondering if the authors had a reason for only trapping and mist netting during a specific time of the year. Presumably, these animals are pretty active throughout much of the year outside of the summer season. Would other tick species or life stages be found on these animals during the spring or fall, thus possibly contributing other pathogens at other time points of the year? Are the models and statistics held true outside of the months where trapping and mist netting occurred? I saw that the authors conducted tick dragging May-October, but tick dragging does not seem to result in a high diversity of tick species compared to ticks found on animals and is not a replacement for trapping or mist-netting animals to find ticks.

We aimed to sample during the main period of the year when hosts are abundant. For birds, that is from early summer after birds arrive after their annual northward migration, to late summer after which they leave on their annual southward migration. Small mammals are active throughout the summer season and are particularly abundant in our area in late summer, when the juveniles of the year are also present in the population (Jutras, 2005). Since it was not logistically possible for us to deploy the effort for trapping and mistnetting at the same time, we placed the bird and rodent capture periods in sequence, during these periods of hosts activity. The period during which sampling occurred also covered the main period of immature tick activity in this site that we have been studying for the past 15 years (e.g. Bouchard et al., 2011; Dumas et al., 2022; Ripoche et al., 2018). This is clarified (Page 7, Lines 149-150). Furthermore, during our studies we have found other tick species that can transmit B. burgdorferi (I. muris, I. marxi and I. angustus), but as in our study here (where the only ticks of other species found were two I. muris ticks), they are very uncommon compared to I. scapularis, and it is unlikely they are contributing greatly to pathogen transmission on the site. Consequently, it is also very unlikely that there are patterns of tick parasitism of hosts occurring outside the sampling period that would impact the generalisability of the results during the sampling period.

2.2. Interestingly, the authors did not detect any pathogens in the bird biopsies, but found that the larval ticks were infected with pathogens. While the Discussion noted that the birds can possibly clear the infection, this still does not explain how the ticks were infected with the pathogen if the bird is cleared of infection by the time it was sampled. Larval ticks only feed for a few days, so is it possible that birds are clearing pathogens before the larvae are replete? Do the authors have data showing that more fully-engorged ticks have a higher pathogen load compared to a tick that just started feeding? This could provide strong evidence for their theory, where as a host is clearing the infection, ticks that fed sooner would be expected to have a higher pathogen load compared to those that started feeding later.

We added to the discussion possible explanations for the lack of positive bird biopsy samples (Page 48, Lines 645-658). This is not to be confused with the explanation that some birds might clear the infection from ticks that have fed on them. We believe this to be an immune mechanism specific to certain species only, a theory we advance since, for these few species, we found engorged larvae and nymphs on the birds, but none were infected (see Discussion). It would be interesting to study how birds of different species may acquire immunity to B. burgdorferi, resulting in changes in host-to-tick transmission efficiency, but we don’t have the detailed data for this in our studies to date.

2.3. Furthermore, I do notice that more emphasis is given to the data on birds, while white-footed mice and other small mammals do not seem to garner the same level of attention. While I do agree that birds have been understudied as a tick host/tick-borne pathogen reservoir, I still think that the small mammal component is informative. If your emphasis is on birds, then the small mammal component should be removed to make the paper more focused and succinct. I personally think that both should be kept in the paper, but they should also be equally represented.

We have made some choices in the presentation of the data (e.g., Table 2 dedicated to birds), due to the great diversity of bird species encountered in our sample. However, all data for mice and other small mammals are also presented in the Results section. Also, it is true that the results for birds are discussed extensively in the article. This is due to the greater novelty of these results and the objective of the study to compare the role of these alternative hosts to the already recognized role of rodents such as the white-footed mouse. Nevertheless, we have taken care to better represent the results of the two host groups through additions throughout the paper (see Abstract, Introduction and Discussion). 

Minor Comments: 

2.4. Line 33: Add sample sizes into the abstract.

Added.

2.5. Line 36: There doesn’t seem to be any information on the small mammal infestation and infection data or the reservoir potential value in the abstract. Was there a reason to exclude it or could you add it to the abstract to make it more representative of your data?

Added.

2.6. Lines 54-55: Diseases should be lowercased.

The word “Diseases” was not present in lines 54-55. Correction was made though from “relapsing Borreliosis caused by Borrelia miyamotoi” to “Borrelia miyamotoi disease” accordingly to CDC nomenclature (page 3, line 56).

2.7. Lines 54-55: The last part of this statement (“…caused by Borrelia miyamotoi”) makes it sound like all of the diseases you listed are caused by B. miyamotoi. Maybe instead you could list the relevant pathogens that cause the disease in parentheses after the disease is mentioned.

Modification made (see previous point).

2.8. Line 88 (and elsewhere): I think common bird names should be lowercased unless they are named after someone or a location. In addition, you might want to consider putting the scientific names of birds in parentheses after the common name is listed the first time. This was done for the small mammals in the Introduction and Results sections, so I think it should also be done for the birds for consistency.

Scientific names have been added to the first mention of each bird species. However, common bird names were kept capitalised, in accordance with the guidelines of the American Ornithological Society (https://americanornithology.org/nacc/guidelines-for-english-bird-names/). 

2.9. Lines 127-130: What were the methods for tick dragging and storing the ticks before pathogen testing? Are they included in another reference? If so, the reference should be added. 

Reference added.

Figure 1 is also listed in the paragraph, but I don’t see the caption for it anywhere in the review packet.

The caption for Figure 1 is in the manuscript text, next to the paragraph where it is first mentioned, in accordance with PLOS ONE submission guidelines “Figure captions must be inserted in the text of the manuscript, immediately following the paragraph in which the figure is first cited (read order). Do not include captions as part of the figure files themselves or submit them in a separate document.” https://journals.plos.org/plosone/s/submission-guidelines#loc-style-and-format

2.10. Lines 132-143: I noticed that the Ethics Statement of the review packet included more information about trapping and bird mist netting and this information should be added to the manuscript itself as well.

Added.

2.11. Lines 141-143: Do you think DNA degradation of your sample is possible if the sample is kept in ethanol and at room temperature for long periods of time?

Preservation of samples in ethanol for PCR analyses is known to be sufficient for short-term storage (Carew et al., 2018; Stein et al., 2013). Over the longer term, gradual degradation of DNA does occur, resulting in decreasing concentrations in samples over years of storage (Barnes et al., 2000). However, the detection limit is very low for the RT-PCR techniques we use, and the storage period is not long enough (approximately 1 year, information added in Materials and Methods, section Diagnostic testing for tick-borne pathogens) for DNA concentrations to fall below the detection limit.

2.12. Line 153: When luring the birds to the nets, do you think you might have biased your mist netting results to specific species? Do you suspect that using sound lures would decrease capture success of other bird species that were not used as playback?

The bird species called with playback were based on those detected during the point counts. Thus, playbacks were used equally for all species detected per site. Also, we do not believe that this technique should decrease the capture success of non-target species. On the contrary, several studies have found that in addition to increasing the capture success of target species, the use of playbacks while using mist nests could also favor capture (Hera et al., 2017; Sebastianelli et al., 2020), or response (Møller, 1992) of other species, not targeted by the playback. This is clarified (Page 8; lines 173-175). Indeed, it has been suggested that birds may use song features to make a general assessment of the quality of a site, such as the presence of food resources when the parents are feeding the young (Sebastianelli et al., 2020). Furthermore, since mist netting was conducted during the nesting season, most of the birds have already settled in their territories. The potential effects of interspecific competition already occurred during territory settlement prior to our mist netting sessions and using playback to stimulate a bird's territorial response should not repel its competitors (who would already have chosen territories elsewhere). By the second capture period, playbacks were much less effective, as birds were no longer defending territories.

2.13. Lines 173-174: “… and a subset of up to 30 nymphs per infested host species for nymphs collected from birds.” What about the samples from mice?

We did not test nymphs collected from mice and other small mammals. This is clarified (Page 9, Lines 202-203).

2.14. Line 175: “Unfed larvae from individual hosts…” Do you mean that the larvae were pooled all together for pathogen testing? Or do you mean that you unfed larvae from the same individual/host were pooled together?

Unfed larvae from the same individual host were pooled together. This is clarified (Page 9 Lines 204-205).

Lines 180-182: Gene names should be italicized (23S, msp2, ospA/flab). “spp.” should not be italicized. Add a comma before “respectively.”

Modifications made.

2.15. Lines 182-184: The way the sentence is structured currently, it sounds like you tested samples from 2016 even though the study took place in 2017 and 2018. I would rephrase it so that you say you didn’t test any samples in 2018 since you did not find any positive samples in your first year of study in 2017. 

Modification made. 

That being said, was there a reason why you didn’t test samples from both years? Wouldn’t results from 2017 be more or less independent of what you might find in 2018? In other words, what you find in 2017 is not necessarily representative of what you find in 2018.

In fact, we decided not to take the biopsies at all in 2018 (clarification made; Page 10, lines 213-214). We found our technique to be ineffective for the detection of pathogens in the tissue we collected (please see the explanations added in Discussion; Page 48, Lines 645-658) and avoided taking these quite invasive samples for ethics reasons. 

2.16. Lines 188-194: Instead of Chi-square analyses, would it be more accurate to test for significant differences using a model, especially since you are testing various origins. The origins could be your covariates and the presence/absence of the pathogen could be the outcome of the model.

As we compared only between engorged larvae and questing nymphs (we clarified this in Page 10, Line 223), we do think that the Chi-square tests are appropriate. Had we been comparing amongst multiple sources for the ticks, a logistic regression model would have been more appropriate. 

2.17. Lines 230-231: If you used any R packages for your statistical analyses, I recommend adding and referencing those packages in the manuscript.

All the packages used and versions are now cited and referenced.

2.18. Lines 230: The sentence structure makes it sound like the DHARMa package conducts Moran’s I tests, but I don’t think it’s used for that. In this case, I would break up the sentence into two sentences and for the former, list the package you used for Moran’s I (see previous comment). R packages should also be italicized.

Corrections made.

2.19. Lines 253-254: Is it possible to assume that tick attachment between recaptures could happen, even within 24 hours? If all ticks were removed and you found ticks the next day, would this event be considered independent and could thusly be included in your analyses?

It can certainly be assumed that tick attachment can occur within 24 hours. However, given that immature ticks remain attached to the host for a few days, tick abundance on a particular animal will likely be associated with the time elapsed since the last time it was captured (and had all of its ticks removed by the research team). In other words, the infestation density for a single individual is not independent of each successive recapture event. Furthermore, we believe that this lack of independence will be more pronounced in rodents than in birds. Indeed, rodents from the same site could be recaptured during the same week, whereas for birds, successive recapture periods at the same sites were spaced several weeks apart. So, considering recaptures in the models would at least have required the addition of random effects at the host individual level. We explored this possibility but given that we already had a random effect for the capture site (necessary to account for spatial dependence and repeated measurements at sites), adding a second random effect made the models too complex and caused convergence problems. We therefore decided to exclude recaptures, in order to keep the models functional and avoid bias. 

2.20. Line 261: “The white-footed mouse…”

Correction made.

2.21. Line 317: “Ticks were found on 245 (28.86%) birds, with the majority of these ticks…”

Correction made.

2.22. Table 2: I may have missed this in the manuscript, but why are some of the densities not calculated for some birds? They either have a blank or a dash in them, but multiple birds were captured. Would it also be possible to have a similar table for the small mammal captures to succinctly describe those results as well?

Because densities were obtained from point counts, not from captures. So, in the case of birds for which we have captures but no density estimates, that means we could not detect these species during the point counts. We are therefore unable to provide density estimates in these cases.

2.23. Line 341: Similar to a previous comment I made, were unfed ticks tested in pools per host? Or were all unfed ticks pooled together? I’m assuming the former, but this should be clarified here and in the main text of the manuscript.

Unfed ticks were tested in pools per host. This is clarified (Page 30; line 386).

2.24. Lines 341-342: I’m not sure if this is PLOS One’s guidelines, but should this information go with the title as a caption? I feel like I would have understood the table better had I known this information before reading the table.

We think this information is part of the table legend and thus we placed it below, following PLOS One’s guidelines: “Tables require a label (e.g., “Table 1”) and brief descriptive title to be placed above the table. Place legends, footnotes, and other text below the table.“

2.25. Lines 344-346: The sentence sounds a little awkward towards the end, so I would suggest something like the following: We tested 2210 questing nymphs (63.52% of the total questing nymphs collected) and found an overall prevalence of 9.10% for B. burgdorferi across the three years of the study.

Modification made.

2.26. Line 349: “We tested all 2257 I. scapularis feeding larvae…”

Correction made.

2.27. Lines 370-371: What did open-habitat bird species carry more of? Ticks overall? Or certain life stages of ticks?

Larvae. Clarification added.

2.28. Lines 370-372: The sentence is a little confusing, but based on my interpretation of the sentence, I think the sentence should be broken up into two separate sentences. One sentence compares open-habitat bird species and forest species, and the second sentence describes ground-nesting species and species that nest in trees and shrubs.

Modification made.

2.29. Lines 376-378: “For mice, sex of the host was the only intrinsic factor to significantly affect the number of ticks infesting hosts, where males…”

Modification made.

2.30. Table 4: Are some of the columns misaligned? Things like “Intercept,” “Predicted density of questing larvae,” and “Number of engorged larvae” look like they should be aligned with the rest of the table. I would also suggest moving the Infectivity Models to a separate table. When you reference them in the paper, it can get a little confusing as to which table you are referencing. I also almost missed the “Infectivity models” header when reading the table the first time. So either make two separate tables or make the header more noticeable.

The table has been reformatted for clarity but kept in one table for conciseness. The headings identifying each model have been made more easily noticeable. 

Line 405: “… but captured insufficient animals…”

Correction made.

2.31. Table 5: I think it might look a little better if you somehow moved the family names and associated data up as a subheader and then list all of the bird species under that family. Then you have another subheader with the next family and list all those individuals under that family and so on and so forth.

Modification made.

2.32. Lines 430-432: This was only briefly mentioned in the Results and B. miyamotoi results were combined for both small mammals and birds – I think finding the first report of B. miyamotoi in larvae from birds is super neat and should be emphasized in a separate sentence in the Results, if possible. (…)

Added (Page 31, Lines 400-403).

(…) That being said, to make the jump that B. miyamotoi circulates in bird populations at the study site might be a reach considering that none of the bird biopsies came up positive for the pathogen.

We have removed this part of the sentence. Please see, however, the previous response about bird biopsies (#1.3, 2.2, 2.16) and the fuller elaboration of our thinking about the pathogens detected from feeding larvae and the possible implications for their circulation in local host populations in the discussion.

2.33. Line 434: “most prevalent tick-borne pathogen, B. burgdorferi.”

Modification made.

2.34. Lines 461-465: This sentence can probably be separated into two sentences.

Modification made.

2.35. Lines 477-478: Again, can we really say this if the birds tested negative? Based on the explanation in lines 484-487, it sounds like birds can clear and infection during a blood meal, but like my previous comment mentioned, larval ticks only feed for a few days at a time. Are there references that note how quickly birds can clear infection and if this differs per species?

Please see previous answers in #1.3 and #2.2. 

2.36. Lines 504-506: I actually thought B. burgdorferi seemed pretty common in your ticks from birds, with infectivity rates between 11-20%. Even if the birds were not positive themselves, the ticks coming from the birds show that the pathogen is pretty prevalent.

Modification made.

2.37. Lines 512-515: I think “and” is the wrong transition word. Maybe using “therefore” or “and thusly” might be better.

Modification made.

2.38. Lines 520-525: This entire sentence is rather wordy and gets long-winded when trying to read it – could there be a way to shorten it or to break it up into a couple of sentences?

Modification made.

2.39. Lines 534-536: Do you happen to have a reference supporting the claim about different energy balances between male and female mice and how it contributes to different immune suppression mechanisms?

Reference added.

2.40. References: Some species/group names should be properly formatted (italicized, capitalized, etc.). See references 28, 35, and 64. Those were the ones I quickly found, but another review might be a good idea.

Corrections made. 

Comments by Reviewer #3: 

3.1. Line 174-175 I assume engorgement was determined visually

Yes. Added clarifications.

3.2. Line 271-276 Were the point counts conducted both visually and aurally? How were the bird calls identified? Were recordings made and compared to sonograms or did someone identify the birds by ear? Were the point counts conducted at the same time of day each time or at different times?

Added clarifications.

3.3. I noticed that occasionally there would be long, cumbersome sentences where the impact of the sentence would be enhanced by breaking it up into multiple sentences. For example lines 42-45, 107-112, 460-465, 591-595.

Modifications made.

3.4. I am not sure if PLoS has an opinion on the capitalization of common names of birds since there is some debate about whether or not that is appropriate. However, please be consistent throughout the manuscript text and tables when capitalizing these names.

Common bird names were capitalised, in accordance with the guidelines of the American Ornithological Society (https://americanornithology.org/nacc/guidelines-for-english-bird-names/). 

3.5. Lines 180 and 182 have gene names that should be italicized

Correction made.

3.6. For R packages, please indicate the version used if available

Added.

3.7. Line 261 "White-footed mouse" is capitalized but all other references in the text have "white" as lower case

Correction made.

3.8. In table 2, please define what "infestation intensity" is.

Definition added.

3.9. Lines 341-342 The numbers that is referred to here is in parentheses whereas the confidence intervals are in brackets

Correction made.

3.10. Line 344 2210 needs a comma: 2,210

Correction made.

3.11. Line 349 2257 and 1226 also need commas: 2,257 and 1,226

Correction made.

3.12. Line 405 capture should be captured

Correction made.

3.13. Table 5 Under the 'Infestation' heading the 'No' needs a period 'No.'

Correction made.

3.14. Lines 446-450 It is not necessary to include this unpublished information and these lines could be removed.

Removed.

3.15. Line 473 White-tailed deer may also be a reservoir for B. miyamotoi and might be worth mentioning here or considering for future studies. Han, S., Hickling, G. J., & Tsao, J. I. (2016). High prevalence of Borrelia miyamotoi among adult blacklegged ticks from white-tailed deer. Emerging infectious diseases, 22(2), 316.

Information added.

3.16. The conclusions were sound and supported by the data but I do have one question concerning a statement made in lines 552-555. The authors state that there were twice the amount of larvae carried by birds compared to white-footed mice but I can't seem to find which data support that in the results.

This was indeed a misinterpretation and the passage has been corrected. 

Other editing step: 

Diagnostic and conversion done, and figure updated.

---

## [Editor Report · Decision Letter 1]

23 Mar 2022

Transmission patterns of tick-borne pathogens among birds and rodents in a forested park in southeastern Canada

PONE-D-21-34840R1

Dear Dr. Dumas,

We’re pleased to inform you that your manuscript has been judged scientifically suitable for publication and will be formally accepted for publication once it meets all outstanding technical requirements.

Kind regards,

Brian Stevenson, Ph.D.

Academic Editor

PLOS ONE
---

## [Editor Report · Acceptance letter]

29 Mar 2022

PONE-D-21-34840R1 

Transmission patterns of tick-borne pathogens among birds and rodents in a forested park in southeastern Canada 

Dear Dr. Dumas:

I'm pleased to inform you that your manuscript has been deemed suitable for publication in PLOS ONE. Congratulations! Your manuscript is now with our production department. 

Kind regards, 

on behalf of

Prof. Brian Stevenson 

Academic Editor

PLOS ONE